# Pseudomonas aeruginosa PA14 produces R-bodies, extendable protein polymers with roles in host colonization and virulence

Bryan Wang[1], Yu-Cheng Lin[1], Alejandro Vasquez-Rifo [2], Jeanyoung Jo [1], Alexa Price-Whelan [1], Shujuan Tao McDonald[1,3], Lewis M. Brown[1,3], Christian Sieben[4,5] & Lars E. P. Dietrich [1✉]

R-bodies are long, extendable protein polymers formed in the cytoplasm of some bacteria; they are best known for their role in killing of paramecia by bacterial endosymbionts. *Pseudomonas aeruginosa* PA14, an opportunistic pathogen of diverse hosts, contains genes (referred to as the *reb* cluster) with potential to confer production of R-bodies and that have been implicated in virulence. Here, we show that products of the PA14 *reb* cluster associate with R-bodies and control stochastic expression of R-body structural genes. PA14 expresses *reb* genes during colonization of plant and nematode hosts, and R-body production is required for full virulence in nematodes. Analyses of nematode ribosome content and immune response indicate that *P. aeruginosa* R-bodies act via a mechanism involving ribosome cleavage and translational inhibition. Our observations provide insight into the biology of R-body production and its consequences during *P. aeruginosa* infection.

[1] Department of Biological Sciences, Columbia University, New York, NY, USA. [2] Program in Molecular Medicine, University of Massachusetts Medical School, Worcester, MA, USA. [3] Quantitative Proteomics and Metabolomics Center, Columbia University, New York, NY, USA. [4] Nanoscale Infection Biology Group, Department of Cell Biology, Helmholtz Centre for Infection Research, Braunschweig, Germany. [5] Institute for Genetics, Helmholtz Technische Universität Braunschweig, Braunschweig, Germany. ✉email: LDietrich@columbia.edu

R-bodies are enigmatic biological machines that have been characterized most extensively in *Caedibacter taeniospiralis*, an obligate endosymbiont of paramecia. These structures are composed of ~ 12 kDa monomers that polymerize to form hypercoiled ribbons ~ 0.5 μm in diameter[1]. In response to changes in parameters such as pH, temperature, or ionic strength[2–4], R-body coils extend into 10–20-μm-long needle-like structures (i.e., ~10× the length of the producing cell)[4,5]. In the *Caedibacter-Paramecium* endosymbiosis, tolerant host paramecia shed *C. taeniospiralis* into the environment, where the bacteria can be taken up by neighboring, sensitive paramecia. R-body-mediated lysis is triggered by lysosomal conditions and thought to release an unidentified factor that is toxic to sensitive paramecia. R-body-producing endosymbionts therefore confer onto their host cell a "killing" phenotype and a competitive advantage in a phenomenon that was first described by Tracy M. Sonneborn in 1938[6]. Subsequent genetic studies have implicated three genes–*rebA, B* and *D*–in *C. taeniospiralis* R-body production. RebA, B, and D are all homologous to each other, but RebA and RebB constitute the primary *C. taeniospiralis* R-body components[7–9].

The opportunistic pathogen *Pseudomonas aeruginosa* is a major cause of biofilm-based infections. *reb* gene homologs are found in a substantial fraction (147/241) of *P. aeruginosa* complete strain genomes in the Pseudomonas Genome DB[10], including that of strain PA14. These homologs occur as part of a cluster that includes the gene for the sigma factor FecI2 and targets of FecI2[11], and we refer to this region as the "*reb* cluster". Genes in the *reb* cluster have been identified as contributing to increased virulence when distinct *P. aeruginosa* isolates are compared[12,13]. Notably, the popular model strain PAO1 does not contain a *reb* cluster and this is consistent with its relatively low virulence in a *Caenorhabditis elegans* infection assay. These observations, combined with the known contributions of R-bodies to host damage in other systems[4,14,15], led us to hypothesize that PA14 produces R-bodies that contribute to pathogenicity.

In this report, we provide evidence that PA14 produces R-bodies. We detail characteristics of the PA14 *reb* cluster and its expression, and identify products of the *reb* cluster that form or associate with R-body-like structures inside PA14 cells. We also show that R-body components are produced during interactions with plant and animal hosts. Moreover, we implicate R-body production in PA14-mediated ribosomal injury, a virulence mechanism that yields translational inhibition in the nematode host *C. elegans*[16]. Taken together, these observations identify the R-body as a PA14 virulence factor and raise the possibility that R-body production is a component of the marked strain-dependent variation in pathogenicity observed for *P. aeruginosa*[13,17,18].

## Results

### Phylogenetic analysis of pseudomonad R-body-associated genes and description of the PA14 reb cluster. A previous analysis showed that homologs of *C. taeniospiralis reb* genes are broadly distributed throughout the proteobacterial clade[19]. To investigate the phylogenetic relationships between Reb homologs from proteobacterial genomes, we BLASTed PA14_27640, which is the PA14 protein most similar to *C. taeniospiralis* RebB, against all complete bacterial genomes used for prior analyses conducted by Raymann et al.[19] plus 14 representative pseudomonads. We generated a tree that shows the relationships of all Reb homologs from these proteobacteria (Fig. 1a). In this tree, *C. taeniospiralis* RebA and RebB form a distinct grouping and are closely related to Reb homologs from non-pseudomonad gammaproteobacteria, such as those in the genera *Shewanella* and *Aliivibrio*

(Supplementary Fig. 1). Reb homologs from pseudomonads fall into two groups: (i) one that contains *C. taeniospiralis* RebD and PA14_27685 and for which we have therefore designated its members as "RebD"; and (ii) one that is distinct from the *C. taeniospiralis* Rebs but that includes representatives from many *reb* cluster-containing pseudomonads and chromobacteria, and for which we have therefore designated its members as "RebP". The two *rebP* homologs present in PA14, *PA14_27640* and *PA14_27630*, are predicted to form an operon that we refer to as *rebP1P2*[10].

We examined the *reb* clusters—i.e., chromosomal regions of representative pseudomonad strains that contain *rebD, rebP1*, and *rebP2*— in representative pseudomonad strains and found that there are three additional ORFs that are well-conserved. These include two regulatory genes, *PA14_27700* and *fecI2*, which code for a transcription factor and sigma factor, respectively, and which in a prior study were required for full virulence in a *C. elegans* killing assay[18] (Fig. 1b). We have given *PA14_27700* the designation "*rcgA*" (for *reb* cluster gene A). The third gene, *PA14_27680*, was previously shown (together with *PA14_27700*) to contribute to PA14's enhanced virulence when it was compared to *P. aeruginosa* strain PAO1 in *C. elegans* infection studies[12]. *PA14_27680* is conserved in all pseudomonads that contain at least one *reb* structural gene, suggesting that it may be associated with R-body function and we have therefore named it "*rapA*" (R-body-associated protein A). Although *rebP* homologs are scattered across the phylogenetic tree of the genus *Pseudomonas* (Supplementary Figure 2), homologs of the regulatory genes show strong conservation in strains that contain *rebP* genes, both within the species *P. aeruginosa* (in 146/147 *rebP*-containing genomes; Supplementary Data 1) and among other pseudomonads (Fig. 1b). *PA14_27650, PA14_27660,* and *PA14_27675* do not show homology to any characterized genes and are not conserved in *reb* clusters found in other pseudomonad species (Fig. 1b) or other proteobacteria[19].

### PA14 cells growing in biofilms produce R-bodies. For most bacteria that contain *reb* genes, the ability to produce R-bodies has not been investigated. To determine whether PA14 produces R-bodies, we applied a modified R-body enrichment protocol[20] to biofilms formed by the WT and by Δ*pel*. We included the Δ*pel* mutant in our analysis because it is unable to produce the major PA14 exopolysaccharide and therefore yields biofilms that are more amenable to disruption; we suspected that this might increase recovery of proteins from the biofilm and enhance our ability to detect R-bodies. We performed protein mass spectrometry (MS) and scanning electron microscopy (SEM) on the isolated SDS-insoluble fractions (Fig. 2a). Strikingly, three products of the *reb* cluster—RapA, RebP1 and RebP2—were detected by MS at similar levels and were among the 20 most abundant proteins in our samples (Fig. 2b, Supplementary Data 2). One other uncharacterized protein coded for by the *reb* gene cluster, PA14_27675, was also detected specifically in the SDS-insoluble fractions prepared from Δ*pel* biofilms and at levels two times lower than RebP1 and RebP2 (PA14_27675 was not detected in preparations from WT biofilms) (Fig. 2b). Consistent with the MS results, our SEM analyses revealed the presence of R-bodies in preparations from WT and Δ*pel* biofilms (Fig. 2c, Supplementary Fig. 3). We detected R-bodies in various states ranging from hypercoiled (~450 nm long) to extended (~5.5 μm long) (Fig. 2c–e). To assess whether *rebP1P2* is required for R-body production, we prepared SDS-insoluble fractions from Δ*pel* and Δ*pel*Δ*rebP1P2* biofilms and analyzed 16 independent fields of view for each sample. R-bodies were found in all fields of view captured for preparations from Δ*pel* biofilms, with an average of

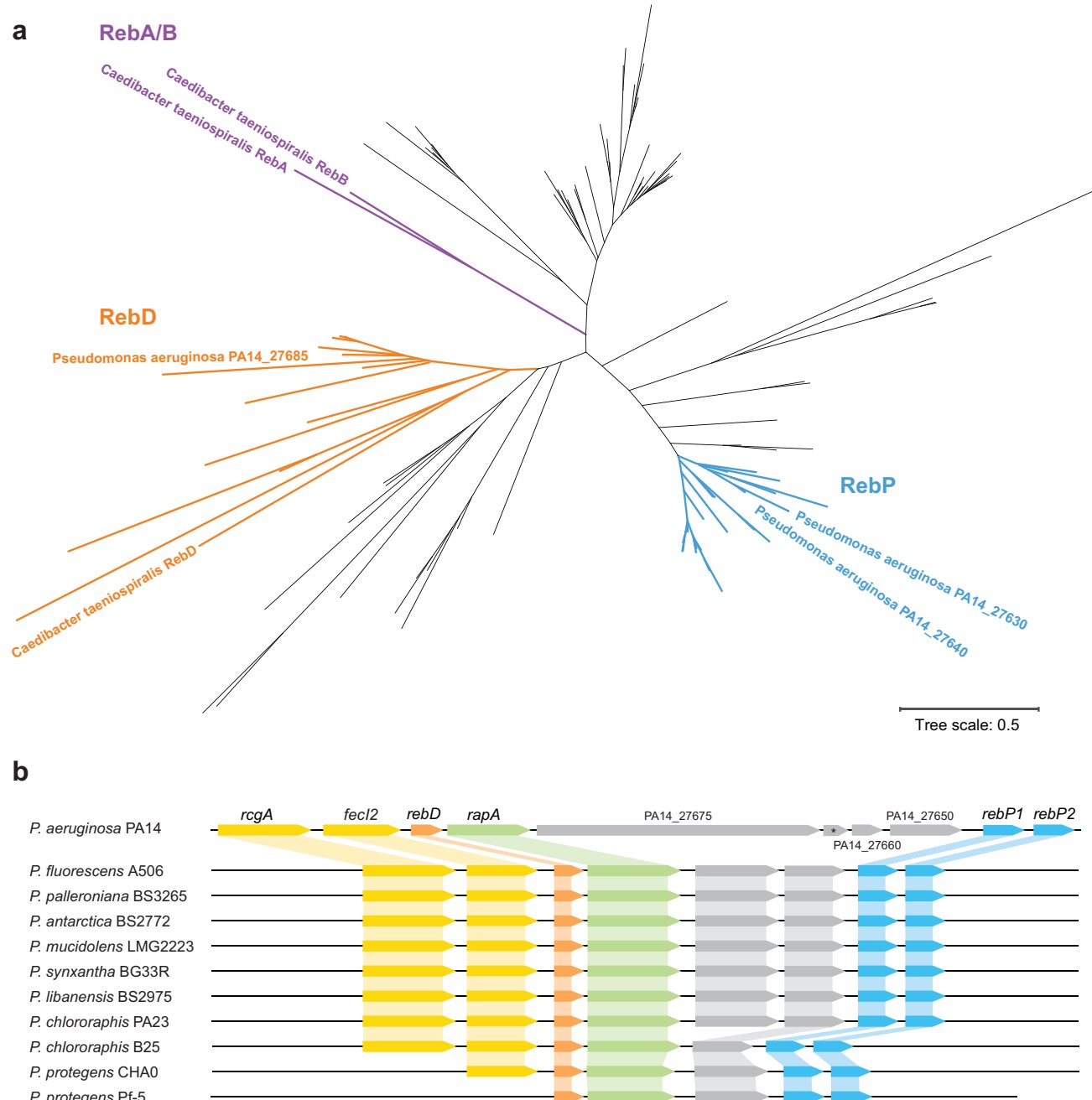

**Fig. 1 Components of the *reb* gene cluster are found in diverse pseudomonads. a** Phylogenetic tree of proteins from 38 bacterial strains that are homologous to *C. taeniospiralis* RebB. The bacterial strains included are those with complete genomes from Fig. 6 of Raymann et al.[19] plus 14 representative pseudomonads (see Supplementary Fig. 1 for the complete annotated tree). *C. taeniospiralis* RebA and RebB are shown in purple. The cluster-containing *C. taeniospiralis* RebD and its *Pseudomonas* homologs is shown in orange. We have assigned the designation "RebP" to Reb homologs that cluster with *P. aeruginosa* PA14_27630 and PA14_27640 (shown in blue). **b** Chromosomal arrangement of genes associated with R-body production in a set of strains that represent all arrangements found in pseudomonads. Regulatory genes are shaded yellow. Genes shaded blue are homologs of *rebP* while genes shaded orange are homologous to *C. taeniospiralis rebD. rapA* (*PA14_27680*) and its homologs, which may code for a novel R-body component, are shaded green. The asterisk denotes a gene that is annotated in BioCyc (*PA14_RS11205*)[57], but not in the Pseudomonas Genome DB[10].

10.8 R-bodies (range = 2–40 R-bodies) seen in each field. R-bodies were not observed in the mutant lacking *rebP1P2* (Supplementary Fig. 3), but were present in amounts similar to the parent strain in images taken of preparations from a complemented strain (Δ*pel*Δ*rebP1P2::rebP1P2*; average of 9.6 R-bodies with a range of 1-30 R-bodies seen in each field). Taken together, our data indicate that R-bodies are produced by PA14 biofilms in a *rebP1P2*-dependent manner.

***rebP1* expression is stochastic and dependent on RcgA and FecI2, in both liquid cultures and biofilms**. Having observed that *rebP1P2* is required for R-body production by PA14 biofilms, we sought to characterize the expression of this locus. We generated a construct that reports expression from the *rebP1* promoter ($P_{rebP1}$) as red fluorescence (mScarlet) and inserted it into a neutral site on the chromosome in WT PA14. We also generated control strains with mScarlet driven by a constitutive promoter[21]

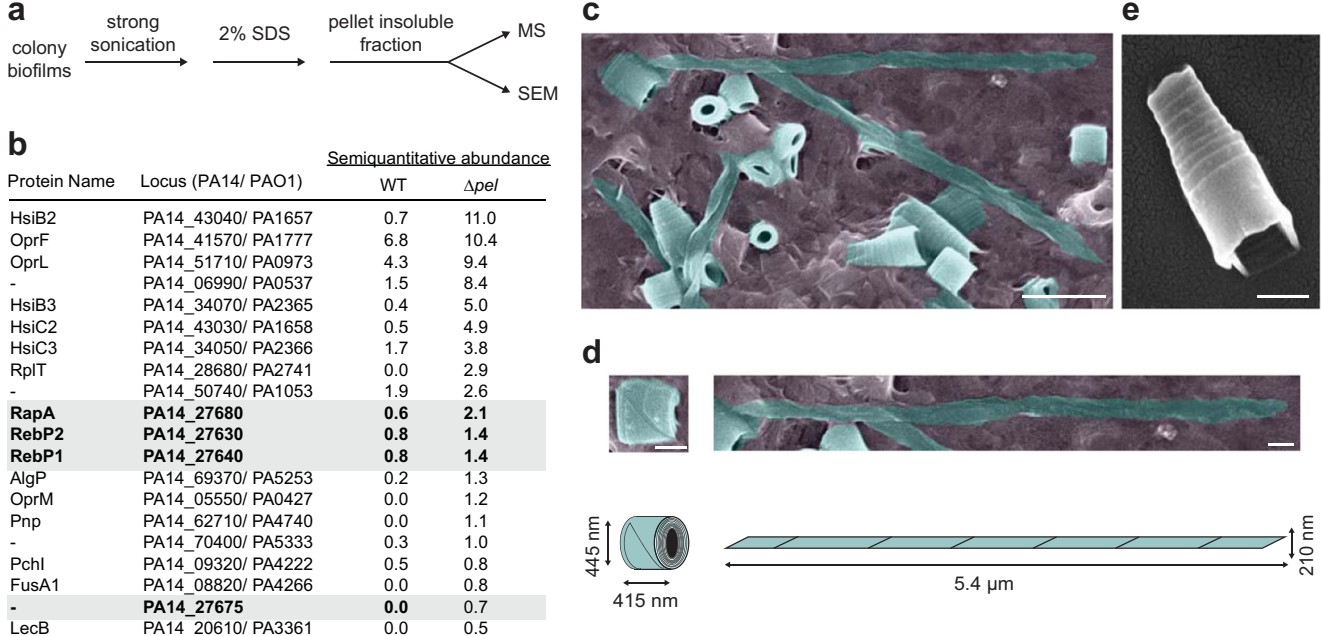

**Fig. 2 R-bodies are produced by *Pseudomonas aeruginosa* PA14 biofilms. a** Schematic showing method for preparation of SDS-insoluble material from PA14 biofilms. **b** Top 20 most abundant proteins detected in the SDS-insoluble, formic acid-solubilized fraction of WT (one replicate) and Δ*pel* (two biological replicates) biofilms by mass spectrometry (MS). **c** Scanning electron micrograph (SEM) image of R-bodies (false-colored turquoise) in the WT sample prepared as shown in (**a**). Scale bar is 1 μm. **d** SEM images, cartoon representations, and dimensions of individual selected R-bodies in fully coiled and fully extended states. Scale bars is 250 nm. **e** SEM image of a single, partially extended R-body. Scale bar is 400 nm. Panels **c–e** scanning electron microscopy are representative of 16 fields of view captured from three experiments.

(P$_{PA1/04/03}$-*mScarlet*) or a promoterless sequence in the WT background. We grew the WT P$_{rebP1}$ and control reporter strains in shaken liquid batch cultures to examine expression. We found that P$_{rebP1}$ showed modest activity compared to that of the constitutive promoter and that it was expressed specifically in the stationary phase of growth (Fig. 3a). To assess the relative expression of these two reporters within single cells, we created a dual reporter strain containing P$_{rebP1}$-*mScarlet* and P$_{PA1/04/03}$-*gfp* and grew it to stationary phase in liquid culture before examination via fluorescence microscopy. This imaging revealed that the low relative fluorescence of the P$_{rebP1}$-*mScarlet* strain (Fig. 3a) was not due to low uniform expression but instead to stochastic expression from the *rebP1* promoter, with mScarlet fluorescence visible in just 0.26% of the cell population (Fig. 3b).

Based on the conservation of *rcgA* and *fecI2* in *Pseudomonas reb* clusters (Fig. 1b; Supplementary Fig. 1; Supplementary Data 1), we reasoned that the encoded transcriptional regulators control expression from P$_{rebP1}$. To test this, we moved the P$_{rebP1}$-*mScarlet* construct into the PA14 Δ*rcgA* and Δ*fecI2* mutant backgrounds. Deletion of one or both of these putative regulatory genes resulted in abrogation of P$_{rebP1}$-driven expression, indicating that RcgA and FecI2 are both required for this activity. Complementation of these genes back into their endogenous loci restored P$_{rebP1}$-driven expression (Fig. 3c, d and Supplementary Fig. 4).

Because biofilm development leads to the formation of steep chemical gradients that can affect gene expression patterns along the *z* axis[22], we chose to visualize P$_{rebP1}$ activity across colony depth. To do this, we employed a thin-sectioning protocol to prepare vertical sections taken from the center region of 3-day-old colony biofilms[23]. Confocal microscopy of the dual reporter (P$_{rebP1}$-*mScarlet* P$_{PA1/04/03}$-*gfp*) biofilm sections revealed that cells expressing *rebP1* are present in thin striations such that both

bright and dark cells are situated at the same depth (red fluorescence in Fig. 3d). The constitutive reporter (P$_{PA1/04/03}$-*gfp*) showed green fluorescence that is relatively homogenous (Fig. 3d). In agreement with our findings for liquid culture, P$_{rebP1}$-driven expression in biofilms appears to be stochastic. The vertical arrangement of P$_{rebP1}$-active cells may indicate that expression status is heritable during vertical growth in the biofilm.

In many hosts and various infection sites in humans, biofilm formation by pathogens such as *P. aeruginosa* is critical to bacterial colonization and virulence. Biofilm formation also exacerbates the challenge of treating resistant infections. Matrix secretion is a defining feature of biofilms; the matrix functions as the "glue" that holds bacterial cells together and can facilitate attachment to the host[24,25]. Having observed R-body production by PA14 biofilms, we sought to examine whether this activity contributes to biofilm development, matrix production, or fitness during biofilm growth. We examined biofilm development using a standardized colony morphology assay[26,27], in which the dye Congo red is provided in the medium and binds to biofilm matrix. We found that matrix production and biofilm development were unaltered in the Δ*rebP1P2* mutant (Fig. 3e). Furthermore, Δ*rebP1P2* showed no fitness disadvantage when grown in mixed-strain colony biofilms with the WT parent (Fig. 3f).

**RebP1 forms internal rings and co-localizes with RapA.** We next sought to assess intracellular R-body production and detect the presence of these structures in vivo. We engineered a strain that produces a partially labeled population of RebP1 molecules by following an approach used previously to study *C. taeniospiralis* R-bodies[20]. In this strain, a construct containing an N-terminal GFP-RebP1 fusion is inserted on the chromosome after the native *rebP1P2* locus (Fig. 4a), and is expressed under the

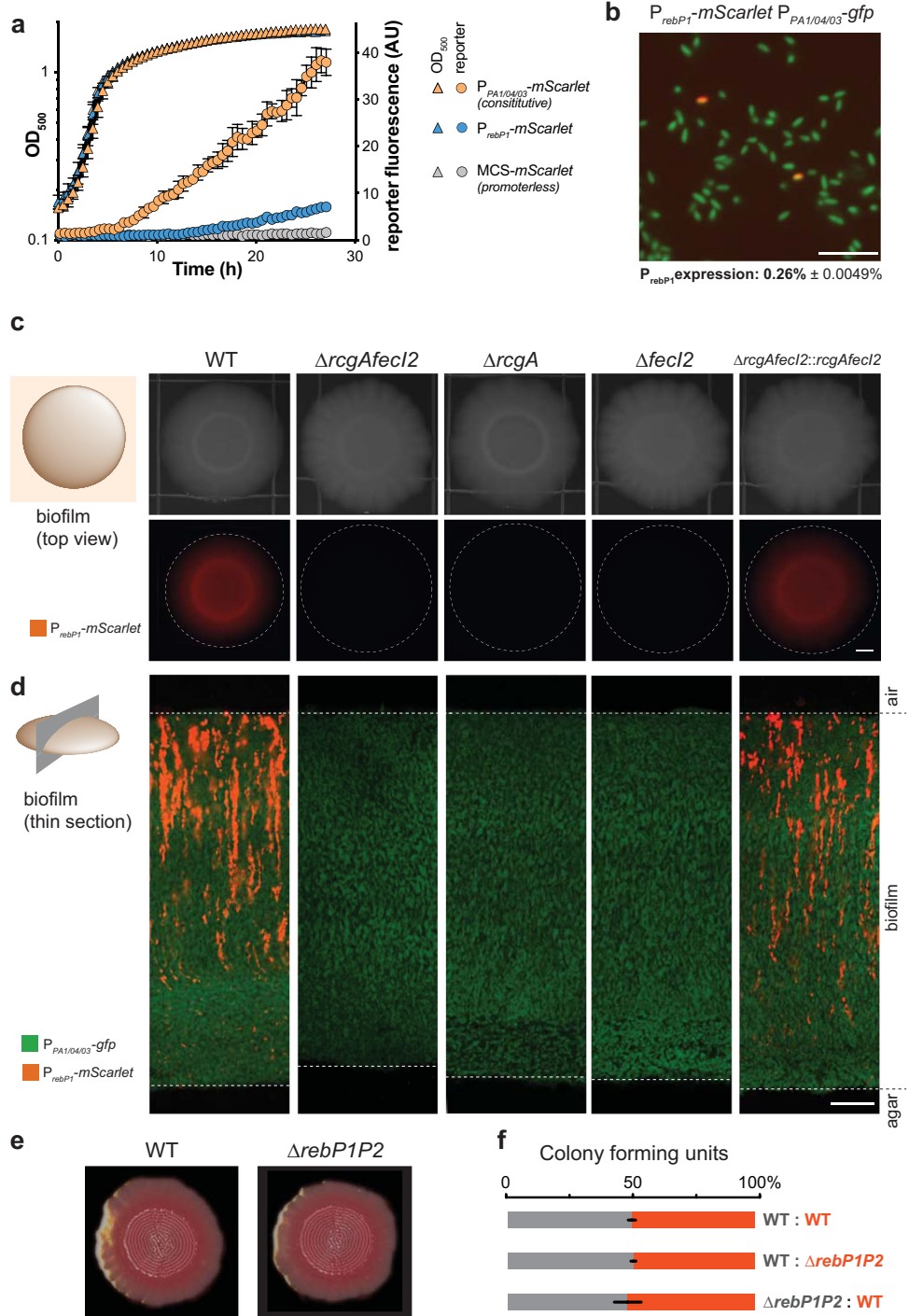

**Fig. 3 Expression of PA14 R-body genes is stochastic and controlled by RcgA and FecI2, and does not confer a competitive advantage during growth in biofilms. a** Reporter strain growth and expression of mScarlet under the control of the *rebP1* promoter (P$_{rebP1}$-*mScarlet*), no promoter (MCS-*mScarlet*), or a constitutive promoter (P$_{PA1/04/03}$-*mScarlet*) in shaken 1% tryptone liquid cultures incubated at 25 °C. $n$ = 6 biological replicates from two experiments. Data are presented as mean values ± SD. **b** Image of fluorescence from PA14 cells containing P$_{rebP1}$ transcriptional reporter construct (P$_{rebP1}$-*mScarlet*) and a constitutive promoter driving *gfp* (P$_{PA1/04/03}$-*gfp*) in shaken 1% tryptone liquid cultures grown at 25 °C for 16 h. Image is representative of six replicates from three experiments. Scale bar is 10 μm. **c** Left: Schematic indicating plane of view, and legend defining fluorescence signal, in right-hand panels. Right: Bright-field and fluorescence microscopy images of 3-day-old biofilms formed by the indicated genotypes, containing the P$_{rebP1}$ transcriptional reporter construct (P$_{rebP1}$-*mScarlet*). Dotted lines demarcate the edges of biofilms. Scale bar is 2 mm. **d** Left: schematic indicating plane of view, and legend defining fluorescence signals, for right-hand panels. Right: representative images of thin sections prepared from 3-day-old biofilms formed by strains containing P$_{rebP1}$-*mScarlet* and constitutively expressing GFP (via P$_{PA1/04/03}$-*gfp*). Dotted lines denote the interfaces with air (top) and agar (bottom). Scale bar is 20 μm. **e** Three-day-old WT and Δ*rebP1P2* biofilms grown on colony morphology medium. Images are representative of nine replicates from three experiments. Scale bar is 4 mm. **f** Relative percentages of colony forming units (CFUs) of each strain obtained from mixed-strain colony biofilms grown for 3 days. Each mixed biofilm contained one unlabeled strain and one constitutively expressing mScarlet. Error bars represent standard deviation of biological triplicates.

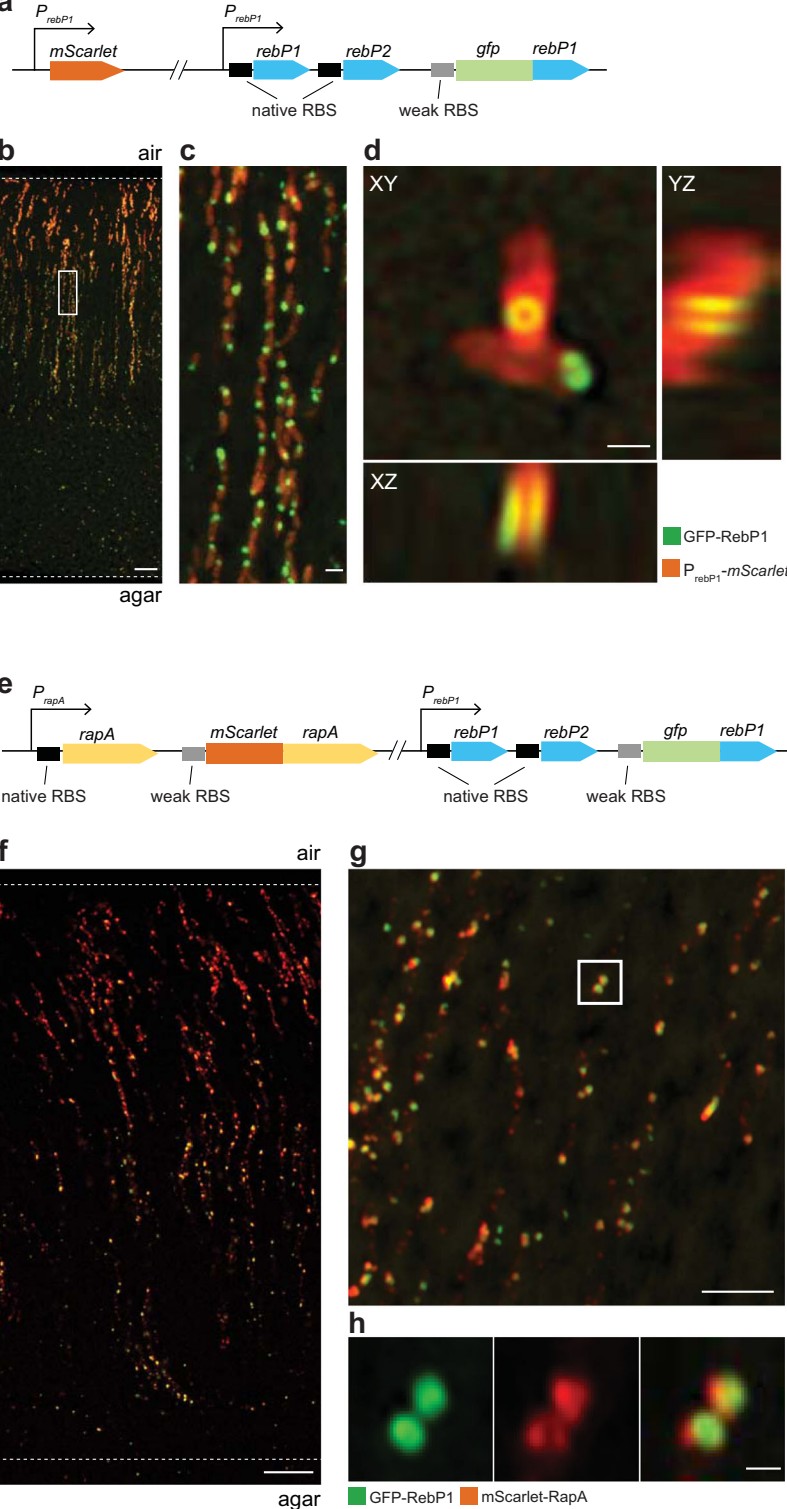

**Fig. 4 RebP1 forms internal rings and co-localizes with RapA. a** Schematic of constructs present in the strain engineered to produce GFP-tagged RebP1, showing the P$_{rebP1}$-*mScarlet* transcriptional reporter and the *gfp-rebP1* translational fusion. **b** Representative confocal image of a thin section prepared from a 3-day-old biofilm formed by the strain indicated. Dotted lines demarcate the edges of the biofilm. Scale bar is 10 μm. **c** Higher-magnification image of the boxed region shows R-bodies within each cell. Scale bar is 1 μm. **d** Zooming in on single cells reveals fluorescently tagged R-bodies forming a characteristic ring structure. Orthogonal projections reveal the tubular open organization of GFP-RebP1. Scale bar is 500 nm. Note that the non-isotropic resolution slightly deforms the structure along the *z* axis. **e** Schematic of constructs present in the strain engineered to produce GFP-tagged RebP1 and mScarlet-tagged RapA, showing the *gfp-rebP1* translational fusion and the *mScarlet-rapA* translational fusion. **f** Representative confocal image of a thin section prepared from a 3-day-old biofilm formed by the strain indicated. Dotted lines demarcate the edges of the biofilm. Scale bar is 10 μm. **g** Super-resolution imaging reveals colocalization of mScarlet-RapA with the R-body ring structure formed by GFP-RebP1. Scale bar is 5 μm. **h** Zoom-in of the boxed region shown above. Scale bar is 500 nm. Images are representative of three fields of view from two experiments.

control of a weak ribosomal binding site (BBa_BB0033). This arrangement yields "leaky" expression of the tagged RebP1, which (relative to tagging of the entire RebP1 population) is thought to decrease potential effects of GFP labeling on R-body structure and function. This strain also contains the $P_{rebP1}$-mScarlet transcriptional reporter (Fig. 4a). We grew this strain as a colony biofilm, prepared thin sections, and imaged them by confocal fluorescence microscopy. We once again observed expression of $P_{rebP1}$-mScarlet in vertically arranged striations. GFP-tagged RebP1 formed discrete spots, with one punctum per cell that could not be further resolved using confocal microscopy. We thus turned to super-resolution microscopy (Zeiss Airyscan2[28]), which revealed that RebP1 is organized as a tubular structure with a diameter of ~250 nm in diameter (Fig. 4d). This structure observed in vivo is conformationally similar, and of comparable size, to the contracted structures seen in our SEM images of the SDS-insoluble fractions prepared from biofilm samples (Figs. 4d and 2d).

The results of the MS analysis indicated that RapA is produced at levels similar to those of RebP1 in biofilms (Fig. 2b) leading us to hypothesize that RapA is associated with R-bodies in vivo. To test this, we created a construct for expression of an N-terminal mScarlet-RapA fusion from a weak ribosomal-binding site and inserted it on the chromosome after the native *rapA* gene (Fig. 4e). The resulting strain was also engineered to contain the construct for leaky expression of GFP-RebP1 as described above. We grew this dual-labeled strain as a colony biofilm and prepared thin sections for confocal and super-resolution microscopy. This imaging revealed that RebP1 and RapA co-localize resulting in a Pearson correlation coefficient of ~0.59 (Fig. 4f, g, Supplementary Fig. 5). Consistent with this observation, super-resolution imaging revealed that RapA indeed overlaps to a large degree with RebP1, indicating that they may form a complex and that RapA is associated with R-bodies in vivo (Fig. 4g, h).

**rebP1P2 is expressed and contributes to PA14 colonization in the plant host *Arabidopsis thaliana***. The fact that homologs of the *C. taeniospiralis* R-body structural genes are found in diverse free-living bacteria suggests that their roles extend beyond endosymbiotic interactions[19]. In *C. taeniospiralis* and the plant symbiont *Azorhizobium caulinodans*, R-body production has been linked to host toxicity[4,14,15,29]. *P. aeruginosa* is an extracellular pathogen that colonizes a range of hosts, including plant and nematode models[30,31]. We therefore sought to probe the roles of R-bodies during these interactions. We first examined whether $P_{rebP1}$ is active and whether *rebP1P2* contributes to colonization after inoculation of the model plant *A. thaliana*. We compared bacterial mScarlet expression patterns on *A. thaliana* seedlings that had been infected with strains containing either the $P_{rebP1}$-mScarlet reporter or the constitutive $P_{PA1/04/03}$-mScarlet construct. We found that while constitutive mScarlet production was uniform and fuzzy across colonized regions of *A. thaliana* leaves, $P_{rebP1}$-driven expression was stochastic and restricted to a subset of PA14 cells (Fig. 5a). To test whether R-body production contributes to host colonization, we inoculated *A. thaliana* seedlings with either PA14 WT or Δ*rebP1P2* and quantified the bacteria recovered after 5 days of plant growth. When compared to WT PA14, Δ*rebP1P2* displayed attenuated *A. thaliana* colonization that was restored to WT levels upon complementation of *rebP1P2* (Fig. 5b). Together, these results indicate a potential role for R-bodies in *P. aeruginosa*–plant host interactions.

**R-bodies contribute to PA14 virulence in the nematode host *Caenorhabditis elegans***. Previous studies with PA14 have demonstrated overlap in the sets of virulence factors that act on

plant and animal models[32]. To investigate R-body production and toxicity in an animal host, we used intestinal colonization and pathogenicity assays in the nematode model *C. elegans*. Consistent with our observations in colony biofilms and on *A. thaliana*, we found that worms fed the strain containing the $P_{rebP1}$-mScarlet reporter showed stochastic fluorescence in the intestine, in contrast to the more uniform fluorescence exhibited by the strain containing the constitutive $P_{PA1/04/03}$-mScarlet construct (Fig. 5c). The distinct $P_{rebP1}$-driven expression pattern we observed in the colony biofilm and host contexts may indicate that there is an advantage to restricting R-body production to a subset of cells in the population. To ask whether R-bodies are present during colonization of the intestine, we infected worms with the strain containing constructs for expression of the GFP-RebP1 fusion protein and the transcriptional reporter $P_{rebP1}$-mScarlet. Fluorescence imaging revealed discrete puncta of GFP-labeled RebP1 in cells expressing mScarlet, indicating that these bacteria contain R-bodies (Fig. 5d).

To test whether R-body production contributes to PA14 virulence in *C. elegans*, we quantified percent survival for up to four days after synchronized populations of worms were exposed to PA14 WT and mutant strains. Mutants lacking *rcgA* or *rapA* displayed attenuated virulence, consistent with previously reported results (Fig. 6b). Δ*rebP1P2* also displayed attenuated virulence, which was restored to WT levels upon complementation of *rebP1P2* (Fig. 6a). The reduction in virulence was more pronounced for Δ*rcgA* and Δ*rapA* than it was for Δ*rebP1P2*, suggesting that RcgA and RapA act via RebP-dependent and RebP-independent mechanisms to enhance virulence. Nevertheless these results, together with the results of our genetic and imaging analyses with engineered strains (Figs. 3c, d and 4e–h), indicate that the contributions of these proteins to PA14 virulence arise in part from their roles in inducing *rebP1P2* expression or associating with RebP1/P2, respectively.

**R-body production contributes to translational inhibition in *P. aeruginosa*-infected *C. elegans***. A recent study revealed that ribosome damage—specifically, cleavage of the ribosomal RNA at helix 69 (H69), a highly conserved feature of the decoding center—is a component of PA14 pathogenicity in *C. elegans* and that the regulator RcgA strongly promotes this infection outcome[16]. Ribosome cleavage leads to translational inhibition, which activates a *C. elegans* immune surveillance mechanism mediated by the *zip-2* pathway[33,34]. Based on this work and our observations, we hypothesized that host ribosome cleavage and *zip-2* pathway activation are mediated, at least in part, by RcgA-dependent induction of R-body production. We tested this by first employing a *C. elegans* strain that reports expression of *irg-1*, a target of the *zip-2* pathway, as GFP fluorescence. When *irg-1::gfp* worms were fed Δ*rcgA*, Δ*rapA*, or Δ*rebP1P2*, we observed little or no induction of *irg-1*, while feeding with WT PA14 or the complemented Δ*rebP1P2::rebP1P2* strain yielded strong reporter expression localized to the worm intestine (Fig. 7a, b). These results suggest that these *reb* cluster genes are involved in PA14-effected translational inhibition in *C. elegans*.

Next, we investigated whether this translational inhibition correlated with ribosomal cleavage by analyzing rRNA isolated from *C. elegans* fed WT PA14 or *reb* cluster mutant strains. Consistent with prior results[16], we found that Δ*rcgA* showed a marked defect in induction of *C. elegans* ribosomal cleavage when compared to WT PA14. A similar defect was exhibited by the Δ*rapA* mutant. The Δ*rebP1P2* mutant also showed diminished induction of ribosomal cleavage, though the defect was not as pronounced as that of Δ*rcgA* and Δ*rapA* and normal levels of ribosomal cleavage were restored in the Δ*rebP1P2* complemented

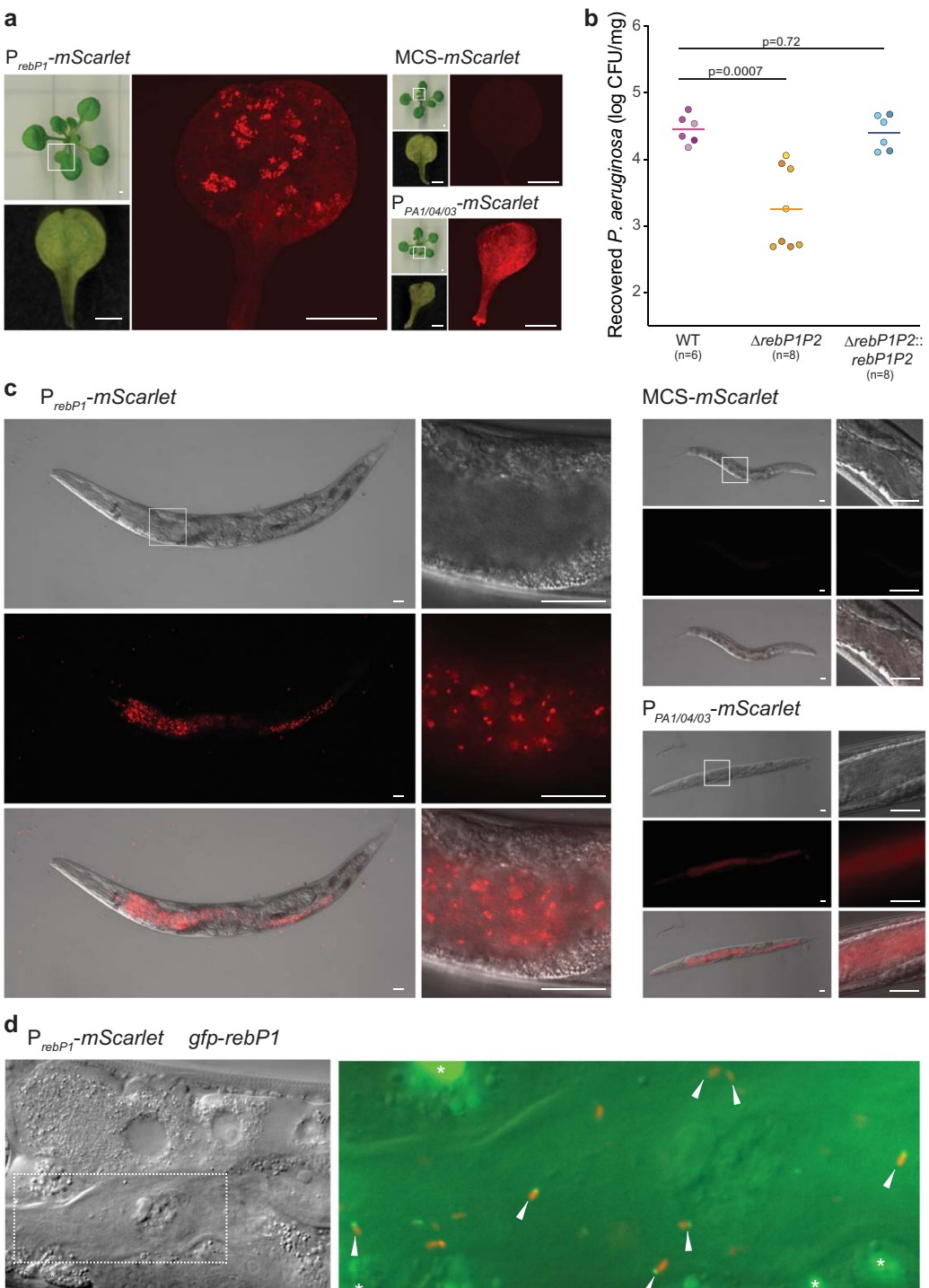

**Fig. 5 R-bodies are produced during host association and contribute to colonization. a** Fluorescent micrographs of *A. thaliana* leaves infected with WT PA14 expressing mScarlet under the control of the *rebP1* promoter (P$_{rebP1}$-*mScarlet*), no promoter (MCS-*mScarlet*), or a constitutive promoter (P$_{PA1/04/03}$-*mScarlet*). Smaller panels show images of the whole seedling and the imaged leaf indicated by the white box. Images are representative of nine biological replicates from three experiments. Scale bars are 1 mm. **b** CFUs obtained from *A. thaliana* seedlings inoculated with the indicated genotypes and incubated for five days, normalized to seedling weight. Each circle represents a biological replicate, with color intensity indicating replicates from the same experiment; colored lines indicate means. *p* values were calculated using unpaired, two-tailed *t*-tests. *n* ≥ 6 biological replicates from three experiments. **c** Images of *C. elegans* fed WT PA14 containing the indicated reporter constructs. Top panels, Nomarski; middle panels, fluorescence; bottom panels, overlay. White boxes indicate regions shown in close-up images. Images are representative of 15 replicates from three experiments. Scale bars are 10 μm. **d** Images of *C. elegans* fed WT PA14 containing indicated transcriptional and translational reporter constructs (construct schematics are shown in Fig. 4a). Left: Nomarski image. Dotted white box indicates region shown in close-up image at right. Right: Nomarski image overlain with red and green fluorescence micrographs. Arrows indicate single PA14 cells in the worm intestine showing both mScarlet and GFP fluorescence. Asterisks (*) indicate autofluorescent granules in the *gfp* channel. Scale bars are 5 μm. Images are representative of 15 replicates from three experiments.

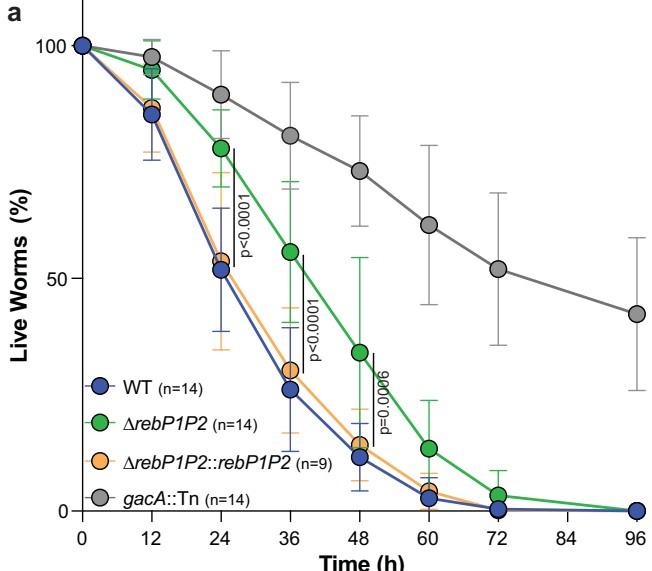

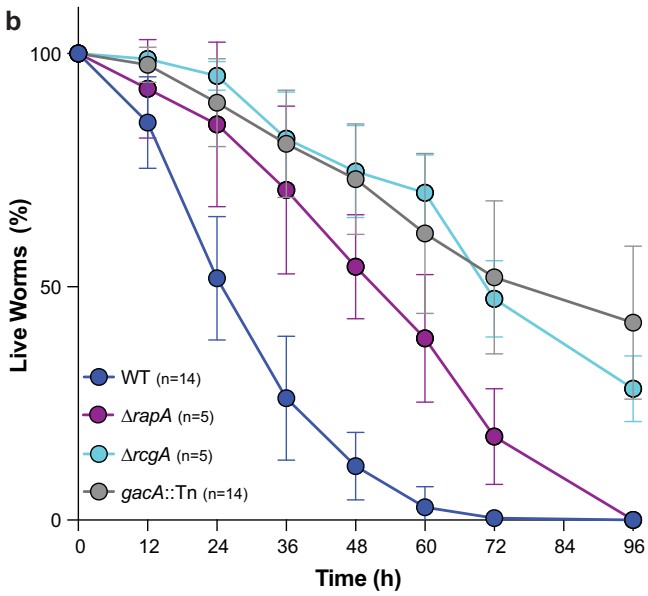

**Fig. 6 *rebP1P2* and genes in the *reb* cluster contribute to host killing. a, b** Killing kinetics of *C. elegans* following exposure to *P. aeruginosa* with the indicated genotypes. The role of *gacA* in *C. elegans* pathogenicity has been described previously[41] and *gacA*::*Tn* therefore serves as a control for defective killing in this assay. Error bars denote standard deviation and p values were calculated using unpaired, two-tailed *t*-tests. $n \geq 5$ biological replicates from four experiments. The n values indicate the number of plates, each containing a PA14 lawn and 30–40 worms. Data are presented as mean values ± SD.

strain (Fig. 7c, d). These results, together with the pathogenicity defects of *reb* cluster gene mutants, (Fig. 6a, b) support the model that R-body-mediated translational inhibition contributes to PA14 virulence in *C. elegans*.

## Discussion

We have identified genes required for R-body production by *P. aeruginosa* PA14. In *C. taeniospiralis*, the proteins RebA, RebB, and RebD have been implicated in R-body production. While these proteins are homologous to each other, RebD is an outlier in that it is shorter than RebA and RebB and in that its expression

is not required for R-body production. PA14 contains one homolog to RebD, and two RebB homologs that we have designated as "RebP1" and "RebP2" due to their distinct phylogenetic grouping. Similar to the case in *C. taeniospiralis*, RebP1 and RebP2 are longer than PA14 RebD, and RebP1 and RebP2 were both detected in SDS-insoluble preparations from PA14 biofilms, while RebD was not. These preparations also contained the product of *rapA*, another *reb* cluster gene that is well-conserved in the pseudomonads, and we found via high-resolution microscopy that RapA is associated with RebP1 inside PA14 cells. Expression of the putative *rebP1P2* operon is regulated by the transcription factor RcgA and sigma factor FecI2. RcgA and FecI2 had previously been associated with virulence[13,18], but their exact roles have eluded definition. We have also provided evidence that *rebP1P2* expression is stochastic and that it can be observed both in independent biofilms and during association with hosts. While expression patterns between these two systems appeared to be similar, an advantage to R-body production was detectable only during host colonization, suggesting that the benefit that R-bodies confer is context-dependent. An interesting possibility is that R-body production is a defense mechanism against grazing protozoa and nematodes, which are also found in soil and aquatic environments that harbor pseudomonads[35–38]. In this context, R-body production could represent a protective strategy that, like type III secretion and rhamnolipid production, has been co-opted into a virulence-factor role[39].

Though R-body production was initially thought to be limited to parasitic bacteria[40], it has since been shown that *reb* gene homologs are found in diverse proteobacteria, including many non-symbionts[19]. Our results indicate that R-bodies made by *P. aeruginosa* PA14 contribute to colonization of a plant host and to virulence in a *C. elegans* model of infection. Aside from *C. taeniospiralis*, and now *P. aeruginosa*, to our knowledge a role for R-bodies in host damage has been observed only for the plant symbiont *A. caulinodans*. The *reb* genes in this plant symbiont are repressed by the regulatory protein PraR under both free-living and host-associated conditions, but deletion of PraR allows R-body production and leads to killing of host plant cells[15]. Because R-body production has not been observed in wild-type *A. caulinodans*, its significance in the context of bacterium-plant symbiosis is unclear[15]. Nevertheless, the fact that the damaging effects of R-bodies have now been described for three divergent bacterial species hints at a general role for these structures in interactions between proteobacteria and eukaryotes.

The *reb* cluster genes *rcgA*, *fecI2*, and *rapA* had previously been shown to contribute to PA14 virulence in *C. elegans*[12,13,18]. *rcgA*, specifically, had also been implicated in the ribosomal cleavage that occurs during *C. elegans* infection[16]. Here, we have shown that the *rebP1P2* locus also contributes to PA14 virulence and ribosomal cleavage, though to a lesser extent than *rcgA* and *rapA*. We have also shown that all three loci play roles in the activation of the *zip-2* pathway in *C. elegans*, and that they contribute to ribosomal cleavage. Combined with our genetic and imaging results suggesting (i) RcgA-dependent activation of P$_{rebP1}$ and (ii) an interaction between RapA and RebP1, our observations in *C. elegans* indicate that RcgA and RapA contribute to virulence via their roles associated with R-body production but also via R-body-independent mechanisms.

*Caedibacter* R-body extension is triggered by low pH and occurs in phagolysosomes of naive paramecia, prompting *Caedibacter* lysis and the release of other toxins within *Caedibacter* cells, thereby killing the eukaryotic predator[4,5]. We speculate that PA14 R-body extension is similarly triggered in a host-specific manner. Cells in the PA14 R-body-producing subpopulation would therefore act as "kamikazes" to release other bacterial products that harm predators or facilitate host colonization and

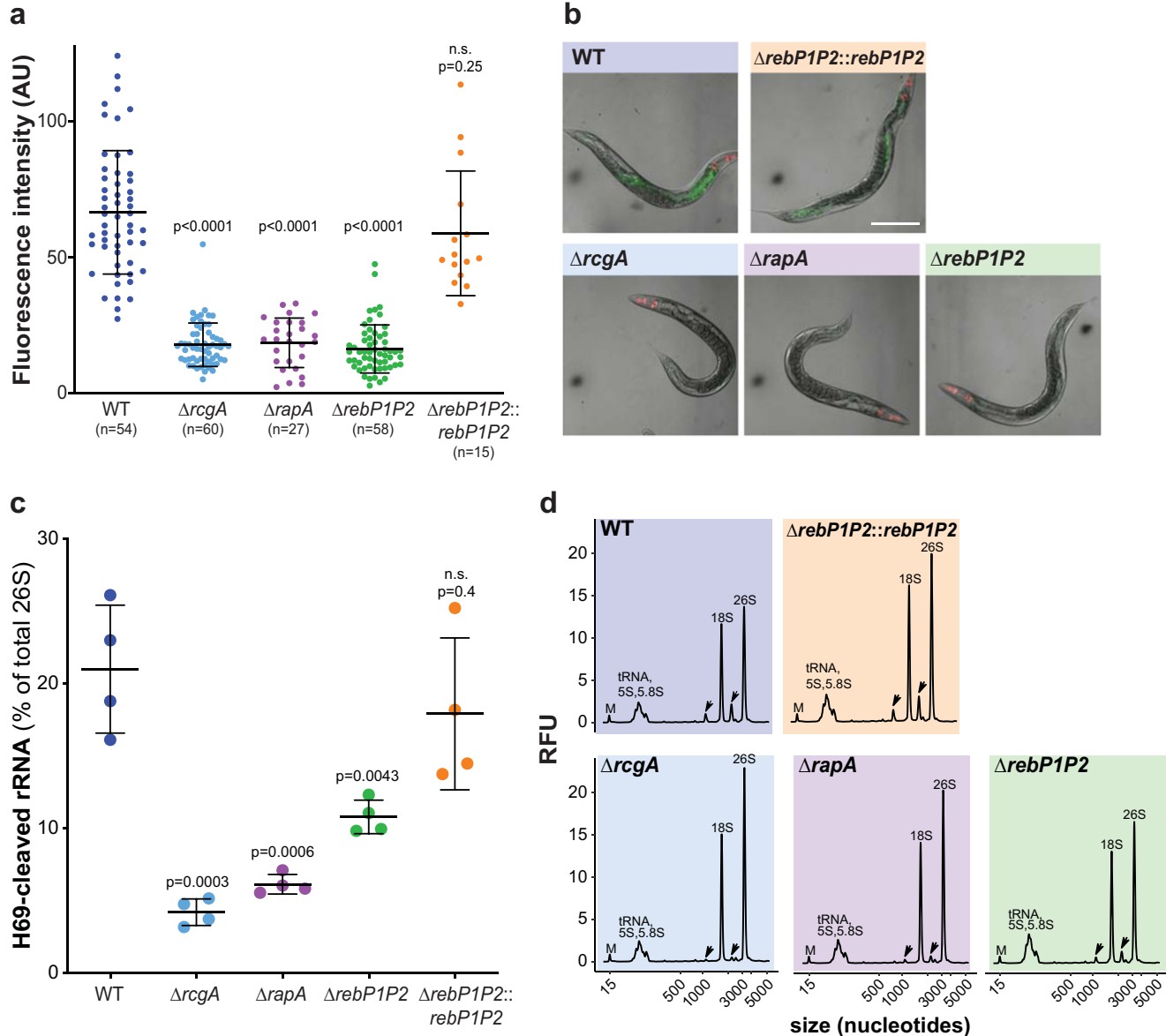

**Fig. 7 Genes in the PA14 *reb* cluster contribute to translational inhibition and ribosome cleavage during *C. elegans* infection. a** Quantification of fluorescence intensity of *irg-1*::*gfp* in the images obtained in **b** using ImageJ/Fiji. *p* values indicated were calculated using unpaired, two-tailed t-tests. n ≥ 15 biological replicates from four experiments. **b** Images of *C. elegans* containing the *irg-1*::*gfp* transgene fed *P. aeruginosa* with the indicated genotypes for 12 h. Each panel shows fluorescence overlayed with Nomarski. Images are representative of ≥15 replicates from four experiments. Scale bar is 100 μm. **c** H69 cleavage levels (as percentage of the total 26S rRNA) in adult worms exposed to the PA14 wild type (WT) or the indicated mutants. *p* values indicated were calculated using unpaired, two-tailed t-tests. *n* = 4 biological replicates from two experiments. Data are presented as mean values ± SD. **d** Total RNA profile of adult worms exposed for 24 h to the indicated *P. aeruginosa* strains. Black arrows denote the H69-cleaved rRNA fragments. RFU relative fluorescence units; "M" indicates a 15-nt marker used in the electrophoretic separation system.

enhance virulence. A model of potential roles for R-bodies in host cell damage is shown in Fig. 8. In this model, either R-body-containing bacteria, or contracted R-bodies themselves, are taken up by host cells via endocytosis. For *C. elegans*, this is supported by the finding that the dynamin gene (*dyn-1*), which is necessary for endocytosis, is integral to the effects of PA14 on *C. elegans* rRNA[16] and *zip-2* pathway activation[16,33]. The low pH of the phagolysosome would trigger R-body extension, either releasing a bacterial enzyme, or activating a host enzyme, that cleaves *C. elegans* rRNA. This ribosomal damage then leads to translational inhibition, activating the *zip-2* pathway and inducing target genes such as *irg-1*. The results of this study therefore indicate that the R-body is a virulence factor and raise the question of whether it

contributes to similar effects in other hosts. More broadly, our findings highlight the possibility that R-body production contributes to the behaviors of diverse free-living bacteria via analogous mechanisms during association with other organisms.

## Methods

**Bacterial strains and growth conditions**. Strains used in this study are listed in Supplementary Table 1. Cultures of *P. aeruginosa* strain UCBPP-PA14 (PA14[41]) were grown in lysogeny broth (LB; 1% tryptone, 1% NaCl, 0.5% yeast extract[42]) in 13 mm × 100 mm culture tubes at 37 °C with shaking at 250 rpm unless otherwise indicated. Biological replicates were inoculated from distinct clonal-source colonies grown on LB + 1.5% agar plates. Overnight precultures were grown for 14–16 h and subcultures were prepared by diluting precultures 1:100 in LB in 13 mm × 100 mm culture tubes and growing at 37 °C with shaking at 250 rpm until mid-

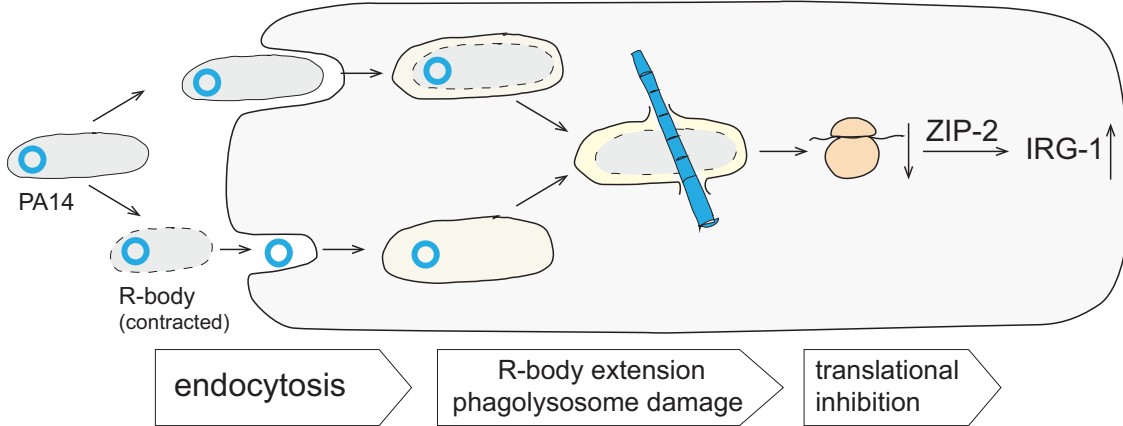

**Fig. 8 Model of R-body action within a host cell.** R-body-containing *P. aeruginosa* cells, or contracted R-bodies themselves, are endocytosed by the host cell (e.g., a *C. elegans* intestinal cell). Conditions of the phagolysosome disrupt the integrity of the bacterial cell wall and/or trigger extension of the R-body. The R-body pierces the phagolysosomal membrane, releasing *P. aeruginosa* cell contents into the host cell cytoplasm. *P. aeruginosa* toxins and/or host responses to lysosomal disruption lead to cleavage of the *C. elegans* ribosome, translational inhibition, and, ultimately, host killing.

exponential phase (OD at 500 nm ~ 0.5). Subcultures were used in experiments unless otherwise noted.

### Construction of markerless deletion and complementation strains[43].

Approximately 1 kb of flanking sequence from each side of the target locus were amplified using the primers listed in Supplementary Table 2 and inserted into pMQ30 through gap repair cloning in *Saccharomyces cerevisiae* InvSc1[44]. Each plasmid, listed in Supplementary Table 2, was transformed into *Escherichia coli* strain UQ950, verified by restriction digests and sequencing, and moved into *P. aeruginosa* PA14 using biparental conjugation via the *E. coli* donor strain BW29427. PA14 single recombinants were selected on LB agar plates containing 100 μg/mL gentamicin. Double recombinants (markerless mutants) were selected on a modified LB medium (containing 10% sucrose and lacking NaCl) and genotypes were confirmed by PCR. Combinatorial mutants were constructed by using single mutants as hosts for biparental conjugation as indicated in Supplementary Table 1.

### Phylogenetic analysis.

All trees were generated with Geneious Prime 2020.2.4 using Muscle for sequence alignment and the neighbor-joining method for tree building (genetic distance model: Jukes-Cantor (Supplementary Fig. 1); Tamura-Nei (Supplementary Fig. 2)). Trees in Fig. 1a and Supplementary Fig. 2 were displayed using the online Interactive Tree Of Life tool[45].

### Construction of PA14 reporter strains.

A transcriptional reporter for the putative *rebP1P2* operon was constructed using primers listed in Supplementary Table 2 to amplify the promoter region (500 bp upstream of *rebP1*), adding an SpeI digest site to the 5′ end and an EcoRI digest site to the 3′ end of the promoter. Purified PCR products were digested and ligated into the multiple cloning site of the pLD3208 vector, upstream of the *mScarlet* coding sequence. This plasmid (pLD3210) was transformed into *E. coli* UQ950, verified by sequencing, and integrated into a neutral site in the PA14 genome using biparental conjugation with *E. coli* S17. PA14 single recombinants were selected on M9 minimal medium agar plates (47.8 mM Na₂HPO₄·7H₂O, 22 mM KH₂PO₄, 8.6 mM NaCl, 18.6 mM NH₄Cl, 1 mM MgSO₄, 0.1 mM CaCl₂, 20 mM sodium citrate, 1.5% agar) containing 70 μg/mL gentamicin. The plasmid backbone was resolved out of PA14 using FLP-FRT recombination by introduction of the pFLP2 plasmid[46] and selected on M9 minimal medium agar plates containing 300 μg/mL carbenicillin and confirmed on LB agar plates without NaCl and modified to contain 10% sucrose. The presence of *mScarlet* in the final clones was confirmed by PCR. To create the strain that constitutively expresses *mScarlet*, the constitutively expressing synthetic *lac* promoter (P_PA1/03/04) was ligated into pLD3208 and mating the resulting plasmid, pLD3433, into the *glmS* locus of the PA14 genome via triparental mating with *E. coli* donor strain BW29427 and *E. coli* helper strain ß2155.

### Colony biofilm morphology assays.

Ten microliters of liquid subcultures was spotted onto 60 mL of colony morphology medium (1% tryptone, 1% agar containing 40 μg/mL Congo red dye and 20 μg/mL Coomassie blue dye) in 100 mm × 100 mm × 15 mm square Petri dishes (LDP D210-16)[27]. Plates were incubated for up to 5 days in the dark at 25 °C with >90% humidity (Percival CU-22L) and imaged daily with an Epson Expression 11000XL scanner. Images shown are representative of at least six biological replicates from three independent experiments.

### Competition mixing assays.

ODs at 500 nm of subcultures were determined in a Synergy 4 plate reader (BioTek) and subcultures were mixed together in a 1:1 ratio of fluorescent, mScarlet-expressing and non-fluorescent cells. Ten microliters of this mixture were spotted onto colony morphology medium and grown as described above. After three days, biofilms were harvested, resuspended in 1 mL of 1% tryptone, and homogenized in an Omni Bead Ruptor 12 bead mill homogenizer for 99 s on the "high" setting. Serial dilutions of homogenized cells were plated onto 1% tryptone + 1.5% agar plates and grown overnight at 37 °C and colony-forming units (CFU) were determined. Fluorescent CFUs were determined by imaging with a Typhoon FLA7000 fluorescent scanner (GE Healthcare).

### Purification of SDS-insoluble biofilm fraction.

To isolate the sodium dodecyl sulfate-insoluble (SDS-insoluble) fraction of biofilms, we used wild-type PA14 and a mutant defective in producing the main exopolysaccharide Pel (Δ*pel*). Biological replicates of 5-day-old colony biofilms (*n* = 20, grown on morphology agar as described above) were harvested from the growth medium with a sterile pipette tip, resuspended in 80 mL of sterile phosphate-buffered saline (PBS), and sonicated for 2 × 10 s on ice with the microtip of a Sonifier 250 (Branson). SDS and β-mercaptoethanol were added to final concentrations of 2% and 5%, respectively, and the sample was nutated on a Nutating Mixer (Fisher Scientific 88 861 043) at room temperature (RT; ~ 25 °C) for 60 min. The sample was then transferred into 1.5-mL microfuge tubes and the insoluble fraction spun down at 16,873 × *g* for 5 min. The supernatant was discarded and the pellets (i.e., the Congo red-binding, SDS-insoluble fraction) were pooled together.

### MS of SDS-insoluble proteins.

The SDS-insoluble pellet was washed three times with Optima water (Fisher Scientific), solubilized in 100 μL of 98% formic acid at room temperature for 1 h, spin-vacuum dried for 1.5 h, washed with 50% methanol and water, and then dissolved in 2% RapiGest (Waters Corp.) with 6 mM DTT. Samples were sonicated, boiled, and cooled. Cysteines were reduced and alkylated before proteins were digested with trypsin[47]. Liquid Chromatography mass/spectrometry (120 min runs) was performed with the Synapt G2 (Waters Corp.) in positive resolution/ion mobility mode with proteins identified with ProteinLynx Global Server (PLGS)[48]. Proteins were also identified and semi-quantitatively measured with a Q Exactive HF (Orbitrap, Thermo Scientific) with Mascot V. 2.5[49]. Identification of the most abundant proteins in these preparations by the Q Exactive HF were confirmed by the qualitative orthogonal method (Synapt G2 ion mobility/PLGS analysis). The reference proteome UP000000653 for *P. aeruginosa* strain PA14 from UniProt (Release 2015_10, 11/10/2015) was used for all database searches. All raw MS data files are publicly available at the MassIVE data repository (https://massive.ucsd.edu).

### Scanning electron microscopy.

The SDS-insoluble pellet was resuspended in pre-fixative solution (2% paraformaldehyde, 2.5% glutaraldehyde, 0.0075% L-lysine in PBS) and nutated at RT for 30 min in the dark. The pellet was then washed in sterile PBS and fixed in 2.5% glutaraldehyde in PBS at RT for 30 min in the dark. Fixed pellets were washed twice in PBS and dehydrated through a series of ethanol washes (25%, 50%, 75%, 95%, 3 × 100% ethanol). Samples were visualized with a Helios NanoLab DualBeam 660 (FEI). Four to ten fields of view per sample were captured at high magnification to screen for the presence of R-bodies per biological replicate. Images shown are representative of 16 fields of view from three independent experiments.

**Liquid culture growth assays**. Biological triplicates of overnight precultures were diluted 1:100 in 200 μL of 1% tryptone in a flat bottom, polystyrene, 96-well plate (Greiner Bio-One 655001) and incubated at 25 °C with continuous shaking on the medium setting in a Biotek Synergy 4 or Biotek Synergy H1 plate reader. The expression of mScarlet was assessed by taking fluorescence readings at excitation and emission wavelengths of 569 nm and 599 nm, respectively, every 30 min for up to 24 h. Growth was assessed by taking OD readings at 500 nm simultaneously with the fluorescence readings.

**Quantification of cells expressing $P_{rebP1}$-mScarlet**. Five microliters of overnight PA14 cultures was mounted on a 2% agarose pad on a glass slide, and imaged at 63× on a Zeiss Axio Imager D1 epifluorescence microscope with an AxioCam MRm. Five independent fields of view were captured at random for each biological replicate. Number of cells in each image was determined using the MicrobeJ plugin in Fiji[50,51].

**Fluorescence visualization in *P. aeruginosa* colony biofilms**. For whole-colony fluorescence imaging, 5 μL of liquid subcultures were spotted onto 1% tryptone, 1% agar and colony biofilms were grown in the dark at 25 °C with >90% humidity (Percival CU-22L). At least three biological replicates of each strain were prepared in this manner. After 3 days, bright-field images and fluorescence images were visualized with a Zeiss Axio Zoom.V16 fluorescence stereo zoom microscope (excitation, 545 nm; emission, 605 nm for imaging of mScarlet; excitation, 488 nm; emission, 509 nm for imaging of GFP).

**Thin sectioning of PA14 colony biofilms[23]**. After 3 days of growth as described above, biofilms were overlaid with 1% agar and sandwiched biofilms were lifted from the bottom layer and fixed overnight in 4% paraformaldehyde in PBS at 25 °C for 24 h in the dark. Fixed biofilms were washed twice in PBS and dehydrated through a series of 60-min ethanol washes (25%, 50%, 70%, 95%, 3 × 100% ethanol) and cleared via three 60-min incubations in Histoclear-II (National Diagnostics); these steps were performed using an STP120 Tissue Processor (Thermo Fisher Scientific). Biofilms were then infiltrated with wax via two separate 2-h washes of 100% paraffin wax (Paraplast Xtra) at 55 °C, and allowed to polymerize overnight at 4 °C. Trimmed blocks were sectioned in 10-μm-thick sections perpendicular to the plane of the biofilm, floated onto water at 42 °C, and collected onto slides. Slides were air-dried overnight, heat-fixed on a hotplate for 1 h at 45 °C, and rehydrated in the reverse order of processing. Rehydrated colonies were immediately mounted in TRIS-buffered DAPI:Fluorogel (Electron Microscopy Sciences) and overlaid with a coverslip. At least three biological replicates of each strain were prepared in this manner. Differential interference contrast, fluorescence confocal, and super-resolution imaging were performed using LSM800 and LSM980 Airyscan2 confocal microscopes (Zeiss). Images were processed using the Zeiss Zen software. Colocalization was quantified using the JACoP plugin for ImageJ[52].

**A. thaliana colonization assays**. *A. thaliana* ecotype Columbia (Col-0) seeds were sterilized using standard bleach protocols[53]. Washed, surface-sterilized seeds were resuspended in 0.1% agar and subjected to 24 h of cold treatment at 4 °C, then plated and germinated on half-strength MS agar medium containing Gamborg vitamins[54]. Seeds were incubated at 22 °C with a 12-h-light/12-h-dark photoperiod (100–150 μE/m²/s; Percival CU-22L) for 3 weeks. *A. thaliana* seedlings were then inoculated with PA14 using a flood inoculation assay[55] and incubated under the same light/dark conditions for five days. For visualization of PA14-infected *A. thaliana*, leaves were excised and *mScarlet* fluorescence was visualized with a Zeiss Axio Zoom.V16 fluorescence stereo zoom microscope. For quantification of PA14 colonization of *A. thaliana*, seedlings were harvested, weighed, and surface-sterilized with 1 mL of 5% $H_2O_2$ in water for 3 min. These seedlings were washed five times with sterile PBS and homogenized in 1 mL of PBS in a bead mill homogenizer on the "high" setting for 99 s. Serial dilutions of homogenized tissue were plated onto LB agar plates containing 4 μg/mL tetracycline and grown overnight at 37 °C to select for *P. aeruginosa*, and CFU counts were quantified.

**C. elegans pathogenicity assays**. Ten microliters of overnight PA14 cultures was spotted onto NGM agar plates[56] and incubated at 24 h at 37 °C, followed by 24 h at 25 °C. Thirty to 35 larval stage 4 (L4) worms were picked onto the PA14-seeded plates and incubated at 25 °C. For visualization of PA14-infected worms, worms were exposed to PA14 for three days, immobilized in 10 mM levamisole in water, mounted on a 2% agarose pad on a glass slide, and imaged at 20× and 63× on a Zeiss Axio Imager D1 epifluorescence microscope with an AxioCam MRm. For pathogenicity killing assays, which were modified from[30], live/dead worms were counted for up to 4 days after plating onto PA14-seeded plates. *unc-44(e362)* worms, which exhibit body movement deficits, were used instead of wild-type to prevent worms from crawling off the plates.

**Measurement of ribosome cleavage levels in *P. aeruginosa* fed *C. elegans***. Standard slow killing (SK) assays[30] were used to measure the effect that PA14 strains have on the level of ribosome cleavage at helix 69[16]. Twenty-five microliters aliquots of overnight bacterial liquid LB culture was plated on SK agar plates. The bacterial

lawn was spread to cover the complete agar surface and to prevent worms from escaping the bacterial lawn. The seeded plates were incubated at 37 °C for 24 h and at 25 °C for 24 h. Synchronized adult *C. elegans* worms were then added to the plates and exposed for 24 h at 25 °C. The worms were collected with M9 buffer and decanted three times in 15-mL tubes to allow digestion of bacteria inside the worms and to separate adults from larval progeny. Worm total RNA was extracted using acid guanidinium thiocyanate phenol-chloroform extraction. A profile of the total RNA was subjected to capillary electrophoresis using a "standard sensitivity" RNA assay in a 5300 Fragment Analyzer instrument (Agilent Technologies). The profile was analyzed using the Prosize software 2.0 (Agilent Technologies) to quantify the relative abundance of distinct ribosomal RNA species.

**Measurement of *irg-1::gfp* activation in *P. aeruginosa* fed *C. elegans***. Twenty-five microliters of overnight PA14 cultures was spread onto NGM agar plates[56] and incubated at 24 h at 37 °C, followed by 24 h at 25 °C. Thirty to 35 young adult worms containing the *irg-1::gfp* transgene (strain AU133) were picked onto the PA14-seeded plates and incubated at 25 °C. For visualization of *irg-1::gfp* activation, worms were exposed to PA14 for 12 h, immobilized in 10 mM levamisole in water, mounted on a 2% agarose pad on a glass slide, and imaged at 20x on a Zeiss Axio Imager D1 epifluorescence microscope with an AxioCam MRm. GFP fluorescence intensity was quantified using ImageJ/Fiji[51]. Using the polygon tool, an ROI is drawn around the worm body for each image and the sum gray value (integrated density) was measured and plotted.

**Reporting summary**. Further information on research design is available in the Nature Research Reporting Summary linked to this article.

## Data availability
All raw MS data files are publicly available at the MassIVE data repository under MassIVE submission # MSV000087657. Plasmids and strains generated during and/or analyzed during the current study are available from the corresponding author on reasonable request. Source data are provided with this paper.

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

## Acknowledgements

This study was supported by NIH/NIAID grant R01AI103369, an NSF CAREER award to L.E.P.D. and NSF GRFP grant DGE-1644869 to B.W. Mass spectrometer acquisition and operations were funded by New York State Stem Cell Science Board (NYSTEM contracts CO2361 and C029159) with matching funds from Columbia University and the Columbia Stem Cell Initiative (L.M.B.). A.V.-R. acknowledges support by a fellowship from the Pew Charitable Trusts. C.S. acknowledges support through the Helmholtz Young Investigator Group Program. We thank Jason Reed (UNC Chapel Hill) for providing *A. thaliana* Col-0 seeds, Hillary Callahan (Barnard College) for technical support in growing seedlings, Martin Chalfie for provision of *C. elegans* unc-44(e362) and Iva Greenwald for technical support with imaging of *C. elegans*. We also thank Zarina Akbary and Allison Hung for technical assistance in cloning and imaging respectively.

## Author contributions

B.W., Y-C.L., and L.E.P.D. conceived and designed the study. Y-C.L. and B.W. generated strains. B.W. and L.E.P.D. conducted bioinformatic and phylogenetic analyses. Y-C.L. and B.W. prepared samples for MS analyses, conducted by S.T.M. and L.M.B. Y-C.L. and B.W. obtained and analyzed SEM images. B.W. grew and imaged biofilms and biofilm thin sections. C.S. carried out super-resolution imaging and analyses. B.W. performed plant colonization and nematode infection and killing assays. B.W. obtained images of nematodes infected with bacterial reporter strains. A.V.-R. carried out assays for cleavage of *C. elegans* rRNA. B.W., J.J., A.P-W. and L.E.P.D. wrote the paper with input from A.V.-R., L.M.B. and C.S.

## Competing interests

The authors declare no competing interests.
