## [Peer Review File · Nature Communications]

REVIEWER COMMENTS

Reviewer #1 (Remarks to the Author):

The authors report the characterization of a cluster of reb homologous genes proposed to be involved in production of R-bodies in a clinical isolate of *P. aeruginosa*, strain PA14. Based on previously published data showing that genes found in the reb cluster play an important role in virulence, the authors hypothesize that R-bodies contribute to bacterial pathogenicity. The authors characterize the PA14 reb cluster and provide evidence of R-body production in PA14 biofilms. Results show that the PA14 rebP1P2 operon, with homology to reb genes from *Caedibacter taeniospiralis*, is required for the production of R-bodies. R-body proteins are detected in SDS-insoluble fractions obtained from PA14 biofilms, and expression of the putative rebP1P2 operon is regulated by transcription factor RcgA and sigma factor FecI2, two proteins previously found to be important for *P. aeruginosa* virulence. The authors also present data showing that rebP1P2 expression in planktonic cells and biofilms is stochastic. Additionally, data show that rebP1P2 is expressed during interactions with the model hosts *Arabidopsis thaliana* and *Caenorhabditis elegans*. Finally, results obtained using the PA14 Δ rebP1P2 mutant in the two different model systems suggest that R-bodies are required to reach wild-type levels of *A. thaliana* seedling colonization and full virulence in a *C. elegans* killing assay.

The findings are novel and they are particularly interesting because they establish a link between R-body production and bacterial pathogenesis in *P. aeruginosa* strain PA14. Moreover, the data suggest that R-body production may account in part for differences in the virulence phenotype observed among *P. aeruginosa* strains.

In addition, the implications of this finding will potentially impact not only our understanding of *P. aeruginosa* virulence mechanisms but also of other Reb-harboring bacterial pathogens. Finally, Reb homologs are present in taxa with very different lifestyles, and the role of R-bodies in nature is very diverse. The findings reported in this manuscript will contribute to expand the current knowledge of mechanisms that involve production of R-bodies.

The authors present convincing evidence that address different aspects of their study, including cluster characterization, expression of reb genes under different conditions, R-body production, and involvement of R-bodies in biofilm development and pathogenesis. The statistical analysis is appropriate; the number of biological replicates, and the number of independent experiments performed is adequate.

Specific Comments:

Page 3, ln 75: The authors should replace ..."the prior analyses conducted by 16 plus 14 representative pseudomonads"...by "the prior analyses conducted by Raymann et al. (2013)".....

Page 3, ln 87-103: There is no mention of the homology of uncharacterized genes PA14_27650, PA14_27660 and PA14_27675 to reb genes in the section describing the PA14 reb cluster. The authors should include data related to those genes if available.

Page 4, ln 108: The authors should include a short sentence describing anticipated results for the MS and SEM analysis of isolated SDS-insoluble fractions obtained from the Δ pel mutant biofilms, and their prediction of the impact that a biofilm that is more susceptible to disruption could have over R-body production and Reb protein abundance. Moreover, the authors should include a short interpretation of the differences between the results obtained from WT and Δ pel mutant biofilms regarding Reb protein abundance and R-body production. There is only a brief description of the data, but no interpretation of the results.

Page 4, ln 112 and page 5, ln 113: The authors write "...-RapA, RebP1, and RebP2-were detected by MS at similar levels and were among the 12 most abundant proteins...". In the legend for Figure 2b they write: "Top 20 most abundant proteins detected in the SDS-insoluble...". I assume that there is an error in ln 113 and it should read 20 instead of 12 proteins. If that's not the case, the authors should explain the criteria for the use of "the 12 most abundant...".

Page 5, In 120: The authors should mention that the Δ rebP1P2 mutant was constructed using a Δ pel mutant background (Supplementary Figure 3), and should explain the rationale behind the decision. The authors should also mention if they observed differences in R-Body production between the WT and Δ pel mutant.

Page 6, In 171-172: The authors should include a reference for the colony morphology assay.

Page 6, In 173: The results obtained from the colony morphology assay indicate that R-body production is not required for biofilm development in strain PA14. However, the authors should consider the possibility that R-body production associated to development of *Pseudomonas* biofilms could have evolved as a defense mechanism against predators. Development of mature biofilm microcolonies that display acute toxicity towards surface-feeding *Rhynchomonas nasuta* has been proposed as an effective mechanisms used by *Pseudomonas* to resist protozoan grazing [Matz C, Bergfeld T, Rice SA, Kjelleberg S. *Environ Microbiol.* 2004 Mar; 6(3):218-26. PMID: 14871206]. This type of defense mechanism could have eventually evolved into virulence mechanisms.

Page 7, In 203: The authors should re-write the last part of the sentence. The results obtained indicate that R-body production is not required for biofilm development, but they do not indicate that "R-body production is specific to the host-associated lifestyle". See the next comment.

Page 8, In 219: The authors should re-write the second part of the sentence that contains "... an advantage to R-body production is detectable only during host colonization, suggesting that R-body function is specific to infection". The data are not sufficient to draw such a generalization. *P. aeruginosa* is a ubiquitous bacteria that lives also in the soil, and R-body production may be used as a defense mechanism against grazing predators.

Page 9, In 253: A parenthesis is missing after the reference for LB.

Page 10, In 302: Were the repetitions for the colony biofilm morphology assay done in independent experiments?

Page 10, In 315: the authors should state that both WT and a PA14 defective strain (Δ pel mutant) were used during the assay.

Reviewer #2 (Remarks to the Author):

In this study, the authors found that *P. aeruginosa* PA14 having the reb genes produces R-bodies in the biofilm forming states, and that the expression of the reb genes requires *rcgA* and *fec2* genes encoding regulatory proteins. Furthermore, they successfully found that the reb genes are partially required for leaf colonization in *A. thaliana* and virulence in nematodes. This third finding is considered to be an important finding for exploring the role of R-body in the relationship between bacteria and hosts.

However, there seems to be no novelty other than this finding. This study seems to have only provided a starting point for clarifying the mechanism of action of R-body in colonization and virulence.

Major points:

(1) Lines 179-204: The promoter activity of PrebP1 was investigated in *Arabidopsis* leaves and nematodes, but it is still unclear whether the bacterial cells actually present there produce R-body. It seems necessary for the authors to confirm the existence of R-body by observing with an electron microscope.

(2) Lines 225-228: This hypothesis is very interesting and important. To prove it, the authors had better analyze the integrity of rRNA of the nematodes infected with the Δ rebP1P2 mutant.

Minor points

(1) lines 84-85: There is no evidence that the rebP genes form operons.

(2) Reference 14 (Lines 53, 192, 234, 235): The paper to be cited should be "Matsuoka, J. et al. Stringent expression control of pathogenic R-body production in legume symbiont *Azorhizobium caulinodans*. mBio 8, e00715-17.

Reviewer #3 (Remarks to the Author):

Wang et al provide evidence of R-body production by *P. aeruginosa* PA14, and show that the rebP1P2 genes are required for this. It was demonstrated that subpopulations of PA14 cells express R-body structural genes during biofilm growth and during colonization of plant and animal hosts. Moreover, R-body production was required for wild-type levels of colonization of *A. thaliana* seedlings and for full virulence in a *C. elegans* killing assay. I think it is an interesting and well performed study, but it may be too preliminary for publication in Nat Comm. Some of the conclusions are drawn on somewhat preliminary results that could be strengthened with additional evidence. Below I list a few concerns for the authors consideration:

1. What is the evidence that the structures shown in figure 2c,d,e are R bodies? Are they composed of RebP1 and RebP2? The procedure for isolating the SDS-insoluble fraction is pretty harsh and could lead to artifacts. The lack of R bodies for the rebP1P2 mutant in figure S3 is not convincing.
2. It seems that the data shown in figure 2b are based on only one experiment with the WT strain and two experiments with the pel mutant. It would be reassuring if the data were based on more experiments.
3. I am not sure what the authors mean in line 160. As far as I can see, the constitutive reporter is not included in figure 3c,d.
4. The role of the rebP1P2 genes in virulence in *C. elegans* is not very pronounced and is limited to a narrow window. The authors should include other virulence models to make the study more convincing.
5. The authors suggest (line 203 and 220) that R bodies specifically evolved for a host-associated lifestyle of *P. aeruginosa*. However, *P. aeruginosa* is generally believed to have evolved as an environmental bacterium.
6. The suggested role of R bodies in ribosomal injury is quite speculative.

REVIEWER COMMENTS

Reviewer #1 (Remarks to the Author):

The authors report the characterization of a cluster of reb homologous genes proposed to be involved in production of R-bodies in a clinical isolate of *P. aeruginosa*, strain PA14. Based on previously published data showing that genes found in the reb cluster play an important role in virulence, the authors hypothesize that R-bodies contribute to bacterial pathogenicity. The authors characterize the PA14 reb cluster and provide evidence of R-body production in PA14 biofilms. Results show that the PA14 rebP1P2 operon, with homology to reb genes from *Caedibacter taeniospiralis*, is required for the production of R-bodies. R-body proteins are detected in SDS-insoluble fractions obtained from PA14 biofilms, and expression of the putative rebP1P2 operon is regulated by transcription factor RcgA and sigma factor Fecl2, two proteins previously found to be important for *P. aeruginosa* virulence. The authors also present data showing that rebP1P2 expression in planktonic cells and biofilms is stochastic. Additionally, data show that rebP1P2 is expressed during interactions with the model hosts *Arabidopsis thaliana* and *Caenorhabditis elegans*. Finally, results obtained using the PA14 Δ rebP1P2 mutant in the two different model systems suggest that R-bodies are required to reach wild-type levels of *A. thaliana* seedling colonization and full virulence in a *C. elegans* killing assay. The findings are novel and they are particularly interesting because they establish a link between R-body production and bacterial pathogenesis in *P. aeruginosa* strain PA14. Moreover, the data suggest that R-body production may account in part for differences in the virulence phenotype observed among *P. aeruginosa* strains. In addition, the implications of this finding will potentially impact not only our understanding of *P. aeruginosa* virulence mechanisms but also of other Reb-harboring bacterial pathogens. Finally, Reb homologs are present in taxa with very different lifestyles, and the role of R-bodies in nature is very diverse. The findings reported in this manuscript will contribute to expand the current knowledge of mechanisms that involve production of R-bodies.

The authors present convincing evidence that address different aspects of their study, including cluster characterization, expression of reb genes under different conditions, R-body production, and involvement of R-bodies in biofilm development and pathogenesis. The statistical analysis is appropriate; the number of biological replicates, and the number of independent experiments performed is adequate.

Thank you very much for sharing your positive impressions of our study, and for providing constructive suggestions and insights that improved the manuscript.

Specific Comments:

Page 3, In 75: The authors should replace ...”the prior analyses conducted by 16 plus 14 representative pseudomonads”...by “the prior analyses conducted by Raymann et al. (2013)”.....

We have corrected this.

Page 3, In 87-103: There is no mention of the homology of uncharacterized genes PA14_27650, PA14_27660 and PA14_27675 to reb genes in the section describing the PA14 reb cluster. The authors should include data related to those genes if available.

We are also intrigued by these genes. We BLASTed PA14_27650, PA14_27660, or PA14_27675 against all complete bacterial genomes used for the prior analyses conducted by Raymann et al. (2013) plus 14 representative pseudomonads and did not find homology to any characterized genes. For this paper, we chose to focus on genes that are conserved in the *reb* cluster, but we have added a note to the main text about the lack of characterized homologs for PA14_27650, PA14_27660, or PA14_27675.

Page 4, In 108: The authors should include a short sentence describing anticipated results for the MS and SEM analysis of isolated SDS-insoluble fractions obtained from the Δpel mutant biofilms, and their prediction of the impact that a biofilm that is more susceptible to disruption could have over R-body production and Reb protein abundance. Moreover, the authors should include a short interpretation of the differences between the results obtained from WT and Δpel mutant biofilms regarding Reb protein abundance and R-body production. There is only a brief description of the data, but no interpretation of the results.

We view the MS experiment as a “starting point” used to screen for the presence of *reb* cluster gene products in *P. aeruginosa* biofilms. We included the Δpel mutant simply because we had observed that it makes a “looser” biofilm that is easier to homogenize--and we thought that we might have a better chance of detecting *reb* cluster gene products in biofilms that were more amenable to disruption. We did not expect the lack of Pel to substantially affect the MS results other than to perhaps increase recovery of proteins from the biofilm matrix and thereby enable us to detect proteins that may have been present at levels below the detection limit in the WT sample.

We agree that it is interesting that the relative amounts of some of the detected proteins are different between WT and Δpel samples, but based on this dataset we cannot say whether these differences are reproducible. Our goal for this study was to test the hypothesis that R-bodies are made by PA14, and the MS results (and subsequent experiments described in the paper) support this. We have clarified this in the text.

Page 4, In 112 and page 5, In 113: The authors write “...-RapA, RebP1, and RebP2-were detected by MS at similar levels and were among the 12 most abundant proteins....”. In the legend for Figure 2b they write: “Top 20 most abundant proteins detected in the SDS-insoluble...”. I assume that there is an error in In 113 and it should read 20 instead of 12 proteins. If that’s not the case, the authors should explain the criteria for the use of “the 12 most abundant....”.

We appreciate the reviewer pointing this out and we have corrected the error.

Page 5, In 120: The authors should mention that the $\Delta rebP1P2$ mutant was constructed using a Δpel mutant background (Supplementary Figure 3), and should explain the rationale behind the decision. The authors should also mention if they observed differences in R-Body production between the WT and Δpel mutant.

We have revised the text to indicate that the SEM to reveal the absence of R-bodies in the $\Delta rebP1P2$ mutant was carried out in the Δpel background.

Page 6, In 171-172: The authors should include a reference for the colony morphology assay.

We have added this reference to the method description.

Page 6, In 173: The results obtained from the colony morphology assay indicate that R-body production is not required for biofilm development in strain PA14. However, the authors should consider the possibility that R-body production associated to development of *Pseudomonas* biofilms could have evolved as a defense mechanism against predators. Development of mature biofilm microcolonies that display acute toxicity towards surface-feeding *Rhynchomonas nasuta* has been proposed as an effective mechanisms used by *Pseudomonas* to resist protozoan grazing [Matz C, Bergfeld T, Rice SA, Kjelleberg S. *Environ Microbiol.* 2004 Mar; 6(3):218-26. PMID: 14871206]. This type of defense mechanism could have eventually evolved into virulence mechanisms.

This is a fascinating idea and we very much appreciate the reviewer bringing this up. We have added this concept to the Discussion text.

Page 7, In 203: The authors should re-write the last part of the sentence. The results obtained indicate that R-body production is not required for biofilm development, but they do not indicate that “R-body production is specific to the host-associated lifestyle”. See the next comment.

We agree with this point and have revised the sentence.

Page 8, In 219: The authors should re-write the second part of the sentence that contains ...” an advantage to R-body production is detectable only during host colonization, suggesting that R-body function is specific to infection”. The data are not sufficient to draw such a generalization. *P. aeruginosa* is a ubiquitous bacteria that lives also in the soil, and R-body production may be used as a defense mechanism against grazing predators.

We have revised the sentence and now include this concept.

Page 9, In 253: A parenthesis is missing after the reference for LB.

We have added this.

Page 10, In 302: Were the repetitions for the colony biofilm morphology assay done in independent experiments?

We have revised the text to indicate that the repetitions were done in independent experiments.

Page 10, In 315: the authors should state that both WT and a PA14 defective strain (Δpel mutant) were used during the assay.

We have added this information.

Reviewer #2 (Remarks to the Author):

In this study, the authors found that *P. aeruginosa* PA14 having the *reb* genes produces R-bodies in the biofilm forming states, and that the expression of the *reb* genes requires *rcgA* and *fec2* genes encoding regulatory proteins. Furthermore, they successfully found that the *reb* genes are partially required for leaf colonization in

A. thaliana and virulence in nematodes. This third finding is considered to be an important finding for exploring the role of R-body in the relationship between bacteria and hosts.

However, there seems to be no novelty other than this finding. This study seems to have only provided a starting point for clarifying the mechanism of action of R-body in colonization and virulence.

We appreciate the reviewer's comment regarding the importance of our findings with respect to *A. thaliana* colonization and nematode virulence. We feel that there is also substantial novelty in the discovery that *P. aeruginosa* produces R-bodies during biofilm formation and in the identification of regulatory genes and proteins that contribute to R-body production. In particular, the finding that the sigma factor FecI2 and the transcription factor RcgA (PA14_27700) control *reb* cluster gene expression enhances our understanding of their contributions to virulence, which had been identified in prior studies but for which the associated mechanisms were unknown. This paper reveals that FecI2 and RcgA control production of R-bodies and that R-bodies are virulence factors.

Though *reb* genes are found in diverse proteobacteria, R-body-related biology and the potential roles of R-bodies in interactions with eukaryotic organisms are understudied. We were interested in the potential for *P. aeruginosa* to produce R-bodies because it is an opportunistic pathogen and a major cause of recalcitrant infections in humans. Our results add *P. aeruginosa* to the (extremely short) list of organisms demonstrated to produce R-bodies, and provide insight into the composition of *P. aeruginosa* R-bodies. They also show that R-body production contributes to host colonization and virulence, suggesting that interfering with this process could enhance treatment strategies. Finally, Reviewer #1 also brought up the intriguing possibility, which we have added to the discussion, that R-body production functions as a defense mechanism against protozoan grazing. This may be relevant for survival of *P. aeruginosa* and other R-body producing proteobacteria in aquatic environments and soil. We feel that together, all of these findings and implications confer substantial novelty.

Major points:

(1) Lines 179-204: The promoter activity of PrebP1 was investigated in Arabidopsis leaves and nematodes, but it is still unclear whether the bacterial cells actually present there produce R-body. It seems necessary for the authors to confirm the existence of R-body by observing with an electron microscope.

To investigate whether R-bodies are produced during host colonization, we engineered a strain that expresses GFP-tagged RebP1 and found that RebP1 forms discrete puncta in PA14, consistent with the model that it is a major structural component of the R-body. One punctum is present in each cell. Super-resolution microscopy revealed that these puncta are actually rings of fluorescence, characteristic of R-bodies and in line with structures observed in SEM images of SDS-insoluble preparations from PA14 biofilms. Imaging of *C. elegans* infected with this engineered strain revealed RebP1 puncta inside bacterial cells visible within the intestine. These results are described in the text and shown in figures 4 and 5 of the revised manuscript.

(2) Lines 225-228: This hypothesis is very interesting and important. To prove it, the authors had better analyze the integrity of rRNA of the nematodes infected with the $\Delta rebP1P2$ mutant.

To address this point, we first used a *C. elegans* strain that reports expression of *irg-1* as green fluorescence. *irg-1* is expressed in response to activation of the *zip-2*-mediated defense response pathway. Vasquez-Rifo et al. (2020) have shown that this pathway is activated by WT PA14 and that this is concomitant with ribosomal injury. We found that infection with WT PA14 led to induction of *irg-1* (as previously reported) and that the $\Delta rapA$ and $\Delta rebP1P2$ mutants showed undetectable or attenuated induction of *irg-1*.

Next, we initiated a collaboration with Alejandro Vasquez-Rifo (now a co-author on the manuscript), who discovered *P. aeruginosa*-induced ribosome degradation in *C. elegans*. We analyzed the rRNA of *C. elegans* infected with WT, $\Delta rebP1P2$, or $\Delta rapA$ and found that worms infected with $\Delta rebP1P2$ or $\Delta rapA$ showed attenuated ribosomal injury relative to the WT.

All of these results are now described in the text and shown in figure 7 of the revised manuscript.

Minor points

(1) lines 84-85: There is no evidence that the rebP genes form operons.

We have revised the text to indicate that we do not know whether *rebP1* and *rebP2* are co-transcribed.

(2) Reference 14 (Lines 53, 192, 234, 235): The paper to be cited should be "Matsuoka, J. et al. Stringent expression control of pathogenic R-body production in legume symbiont *Azorhizobium caulinodans*. mBio 8, e00715-17.

We apologize for this error and have corrected the citation.

Reviewer #3 (Remarks to the Author):

Wang et al provide evidence of R-body production by *P. aeruginosa* PA14, and show that the rebP1P2 genes are required for this. It was demonstrated that subpopulations of PA14 cells express R-body structural genes during biofilm growth and during colonization of plant and animal hosts. Moreover, R-body production was required for wild-type levels of colonization of *A. thaliana* seedlings and for full virulence in a *C. elegans* killing assay. I think it is an interesting and well performed study, but it may be too preliminary for publication in Nat Comm. Some of the conclusions are drawn on somewhat preliminary results that could be strengthened with additional evidence. Below I list a few concerns for the authors consideration:

1. What is the evidence that the structures shown in figure 2c,d,e are R bodies? Are they composed of RebP1 and RebP2? The procedure for isolating the SDS-insoluble fraction is pretty harsh and could lead to artifacts. The lack of R bodies for the rebP1P2 mutant in figure S3 is not convincing.

To address these concerns, we have revised the description of the results shown and performed additional experiments. Evidence that the structures we observed are R-bodies stems from the following experiments and observations:

(i) The structures shown in figure 2c,d,e are morphologically similar to R-bodies that have been prepared from *Caedibacter taeniospiralis* (see, as examples, figure 8 in Pond et al. 1989 and figure 6 in Quackenbush et al. 1983). The *rebP1P2* locus, containing homologs to the structural genes for R-body production in *C. taeniospiralis*, is required for R-body production in PA14. We have updated figure S3, and amended our description of the results shown, to provide more information regarding our SEM analysis. Our conclusion that *rebP1P2* is required for R-body production was based on imaging 16 independent fields of view for preparations from biofilms formed by each strain, and three independent, replicate experiments were performed.

(ii) RebP1 forms ring structures in vivo, and these are comparable in size to R-bodies we observed in the SDS-insoluble preparations by SEM. To investigate whether PA14 forms R-bodies in vivo, we engineered a strain that produces low levels of GFP-tagged RebP1. This strain contains a construct coding for a

GFP-RebP1 fusion under the control of a weak ribosomal binding site. The construct has been chromosomally inserted after *rebP1P2*. As described in our response to Reviewer #2, comment 1, we found that this strain exhibited puncta of GFP fluorescence, with one punctum per cell. We collaborated with Christian Sieben (Helmholtz Centre for Infection Research), who carried out super-resolution analysis of these fluorescent structures and found that they are rings, comparable in size to R-bodies produced by other bacteria and to those we observed by SEM. These results are now described in the text and shown in the new figure 4.

Pond FR, Gibson I, Lalucat J, Quackenbush RL. R-body-producing bacteria. *Microbiol Rev.* 1989 Mar;53(1):25-67.

Quackenbush RL, Burbach JA. Cloning and expression of DNA sequences associated with the killer trait of *Paramecium tetraurelia* stock 47. *Proc Natl Acad Sci U S A.* 1983 Jan;80(1):250-4.

2. It seems that the data shown in figure 2b are based on only one experiment with the WT strain and two experiments with the *pel* mutant. It would be reassuring if the data were based on more experiments.

As outlined in our response to Reviewer #1, comment 3, we view the MS experiments as a “first-pass” approach to testing the hypothesis that R-bodies are made by PA14. The MS results, combined with the observations we made by SEM, supported this hypothesis. The revised manuscript contains new observations providing further evidence that PA14 produces R-bodies, namely the results of super-resolution imaging experiments that show that GFP-labeled RebP1 forms ring structures inside PA14 cells.

3. I am not sure what the authors mean in line 160. As far as I can see, the constitutive reporter is not included in figure 3c,d.

We had mistakenly referred to panel c at this point in the text and we have corrected this. The green fluorescence visible in Fig. 3d indicates expression of *gfp* under the control of the constitutive promoter. We amended the text to clarify this.

4. The role of the *rebP1P2* genes in virulence in *C. elegans* is not very pronounced and is limited to a narrow window. The authors should include other virulence models to make the study more convincing.

A. thaliana and *C. elegans* are well-established *P. aeruginosa* hosts and numerous studies have employed colonization and infection models in these hosts to characterize their interactions with bacterial pathogens. Relevant references are listed below.

The results we obtained for the slow-kill assay are similar to observations made for other genes linked to PA14 virulence in *C. elegans* (see for example Feinbaum et al. 2012). In this revision, we have expanded the *reb* cluster analysis with a deletion of *rapA*, which also showed a defect in the slow-kill assay. We now show these results and other new data, which we have added to the revised manuscript in figures 6 and 7, further supporting a role for *rebP1P2* in virulence and providing insight into physiological effects in the worm.

Rahme LG, Stevens EJ, Wolfort SF, Shao J, Tompkins RG, Ausubel FM. Common virulence factors for bacterial pathogenicity in plants and animals. *Science.* 1995 Jun 30;268(5219):1899-902.

Feinbaum RL, Urbach JM, Liberati NT, Djonovic S, Adonizio A, Carvunis AR, Ausubel FM. Genome-wide identification of *Pseudomonas aeruginosa* virulence-related genes using a *Caenorhabditis elegans* infection model. *PLoS Pathog.* 2012;8(7):e1002813.

Djonović S, Urbach JM, Drenkard E, Bush J, Feinbaum R, Ausubel JL, Traficante D, Risech M, Kocks C, Fischbach MA, Priebe GP, Ausubel FM. Trehalose biosynthesis promotes *Pseudomonas aeruginosa* pathogenicity in plants. *PLoS Pathog.* 2013 Mar;9(3):e1003217.

Kaletsky R, Moore RS, Vrla GD, Parsons LR, Gitai Z, Murphy CT. *C. elegans* interprets bacterial non-coding RNAs to learn pathogenic avoidance. *Nature.* 2020 Oct;586(7829):445-451.

5. The authors suggest (line 203 and 220) that R bodies specifically evolved for a host-associated lifestyle of *P. aeruginosa*. However, *P. aeruginosa* is generally believed to have evolved as an environmental bacterium.

We appreciate the reviewer bringing this up. Reviewer #1 also pointed this out, and made the suggestion that R-bodies could play a role in defending environmental biofilms from protozoan grazing. We have revised the text to remove assertions that R-bodies specifically evolved to play roles in virulence and now include the idea that they protect against grazing, in soil or aquatic environments, in the Discussion section.

6. The suggested role of R bodies in ribosomal injury is quite speculative.

We have conducted experiments to test the hypotheses that *reb* cluster genes are involved in the PA14-triggered cleavage of *C. elegans* ribosomes, and therefore in activating the *zip-2*-mediated defense response pathway, and now provide evidence for this in figure 7. Details are provided in the response to Reviewer 2, comment #2, above.

REVIEWERS' COMMENTS

Reviewer #1 (Remarks to the Author):

I believe that the authors' Response to Referees Letter and the revised manuscript have completely addressed the comments from the reviewers. Therefore, I would like to recommend the manuscript for publication in Nature Communications.

Eliana Drenkard

Below is my original report with my signature

The authors report the characterization of a cluster of reb homologous genes proposed to be involved in production of R-bodies in a clinical isolate of *P. aeruginosa*, strain PA14. Based on previously published data showing that genes found in the reb cluster play an important role in virulence, the authors hypothesize that R-bodies contribute to bacterial pathogenicity. The authors characterize the PA14 reb cluster and provide evidence of R-body production in PA14 biofilms. Results show that the PA14 rebP1P2 operon, with homology to reb genes from *Caedibacter taeniospiralis*, is required for the production of R-bodies. R-body proteins are detected in SDS-insoluble fractions obtained from PA14 biofilms, and expression of the putative rebP1P2 operon is regulated by transcription factor RcgA and sigma factor FecI2, two proteins previously found to be important for *P. aeruginosa* virulence. The authors also present data showing that rebP1P2 expression in planktonic cells and biofilms is stochastic. Additionally, data show that rebP1P2 is expressed during interactions with the model hosts *Arabidopsis thaliana* and *Caenorhabditis elegans*. Finally, results obtained using the PA14 Δ rebP1P2 mutant in the two different model systems suggest that R-bodies are required to reach wild-type levels of *A. thaliana* seedling colonization and full virulence in a *C. elegans* killing assay.

The findings are novel and they are particularly interesting because they establish a link between R-body production and bacterial pathogenesis in *P. aeruginosa* strain PA14. Moreover, the data suggest that R-body production may account in part for differences in the virulence phenotype observed among *P. aeruginosa* strains.

In addition, the implications of this finding will potentially impact not only our understanding of *P. aeruginosa* virulence mechanisms but also of other Reb-harboring bacterial pathogens. Finally, Reb homologs are present in taxa with very different lifestyles, and the role of R-bodies in nature is very diverse. The findings reported in this manuscript will contribute to expand the current knowledge of mechanisms that involve production of R-bodies.

The authors present convincing evidence that address different aspects of their study, including cluster characterization, expression of reb genes under different conditions, R-body production, and involvement of R-bodies in biofilm development and pathogenesis. The statistical analysis is appropriate; the number of biological replicates, and the number of independent experiments performed is adequate.

Specific Comments:

Page 3, ln 75: The authors should replace ..."the prior analyses conducted by 16 plus 14 representative pseudomonads"...by "the prior analyses conducted by Raymann et al. (2013)".....

Page 3, ln 87-103: There is no mention of the homology of uncharacterized genes PA14_27650, PA14_27660 and PA14_27675 to reb genes in the section describing the PA14 reb cluster. The authors should include data related to those genes if available.

Page 4, ln 108: The authors should include a short sentence describing anticipated results for the MS and SEM analysis of isolated SDS-insoluble fractions obtained from the Δ pel mutant biofilms, and their prediction of the impact that a biofilm that is more susceptible to disruption could have over R-body production and Reb protein abundance. Moreover, the authors should include a short

interpretation of the differences between the results obtained from WT and Δpel mutant biofilms regarding Reb protein abundance and R-body production. There is only a brief description of the data, but no interpretation of the results.

Page 4, In 112 and page 5, In 113: The authors write "...-RapA, RebP1, and RebP2-were detected by MS at similar levels and were among the 12 most abundant proteins...". In the legend for Figure 2b they write: "Top 20 most abundant proteins detected in the SDS-insoluble...". I assume that there is an error in In 113 and it should read 20 instead of 12 proteins. If that's not the case, the authors should explain the criteria for the use of "the 12 most abundant...".

Page 5, In 120: The authors should mention that the $\Delta rebP1P2$ mutant was constructed using a Δpel mutant background (Supplementary Figure 3), and should explain the rationale behind the decision. The authors should also mention if they observed differences in R-Body production between the WT and Δpel mutant.

Page 6, In 171-172: The authors should include a reference for the colony morphology assay.

Page 6, In 173: The results obtained from the colony morphology assay indicate that R-body production is not required for biofilm development in strain PA14. However, the authors should consider the possibility that R-body production associated to development of *Pseudomonas* biofilms could have evolved as a defense mechanism against predators. Development of mature biofilm microcolonies that display acute toxicity towards surface-feeding *Rhynchomonas nasuta* has been proposed as an effective mechanisms used by *Pseudomonas* to resist protozoan grazing [Matz C, Bergfeld T, Rice SA, Kjelleberg S. *Environ Microbiol.* 2004 Mar; 6(3):218-26. PMID: 14871206]. This type of defense mechanism could have eventually evolved into virulence mechanisms.

Page 7, In 203: The authors should re-write the last part of the sentence. The results obtained indicate that R-body production is not required for biofilm development, but they do not indicate that "R-body production is specific to the host-associated lifestyle". See the next comment.

Page 8, In 219: The authors should re-write the second part of the sentence that contains "... an advantage to R-body production is detectable only during host colonization, suggesting that R-body function is specific to infection". The data are not sufficient to draw such a generalization. *P. aeruginosa* is a ubiquitous bacteria that lives also in the soil, and R-body production may be used as a defense mechanism against grazing predators.

Page 9, In 253: A parenthesis is missing after the reference for LB.

Page 10, In 302: Were the repetitions for the colony biofilm morphology assay done in independent experiments?

Page 10, In 315: the authors should state that both WT and a PA14 defective strain (Δpel mutant) were used during the assay.

Eliana Drenkard

Reviewer #3 (Remarks to the Author):

I appreciate that the authors have adequately addressed some of my concerns.